

# An aerosol activation metamodel of v1.2.0 of the pyrcel cloud parcel model: Development and offline assessment for use in an aerosol-climate model

Daniel Rothenberg[1] and Chien Wang[1]

[1]Department of Earth, Atmospheric, and Planetary Sciences, Massachusetts Institute of Technology, Cambridge, MA, USA

*Correspondence to:* Daniel Rothenberg (darothen@mit.edu)

**Abstract.** In order to simulate an aerosol indirect effect, most global aerosol-climate models utilize an activation scheme to physically relate the ambient aerosol burden to the droplet number nucleated in newly-formed clouds. While successful in this role, activation schemes are becoming frequently called upon to handle chemically-diverse aerosol populations of ever-increasing complexity. As a result, there is a need to evaluate the performance of existing schemes when interfacing with these
complex aerosol populations and to consider ways to incorporate additional processes within them.

    We describe an emulator of a detailed cloud parcel model which can be used to assess aerosol activation, and compare it with two activation parameterizations used in global aerosol models. The emulator is constructed using a sensitivity analysis approach (polynomial chaos expansion) which reproduces the behavior of the parent parcel model across the full range of aerosol properties simulated by an aerosol-climate model. Using offline, iterative calculations with aerosol fields from the
Community Earth System Model/Model of Aerosols for Research of Climate (CESM/MARC), we identify subsets of aerosol parameters to which diagnosed aerosol activation is most sensitive, and use these to train metamodels including and excluding the influence of giant CCN for coupling with the model. Across the large parameter space used to train them, the metamodels estimate droplet number concentration with a mean relative error of 9.2% for aerosol populations without giant CCN, and 6.9% when including them. Using offline activation calculations with CESM/MARC aerosol fields, the best-performing metamodel
has a mean relative error of 4.6%, which is comparable with the two widely-used activation schemes considered here (which have mean relative errors of 2.9% and 6.7%, respectively). We identify the potential for regional biases to arise when estimating droplet number using different activation schemes, particularly in oceanic regimes where our best-performing emulator tends to over-predict by 7%, whereas the reference activation schemes range in mean relative error from -3% to 7%. In these offline calculations, the metamodels which include the effects of giant CCN are accurate in continental regimes (mean relative error
of 0.3%), but strongly over-estimate droplet number in oceanic regimes by up to 22%, particularly in the Southern Ocean. The biases in cloud droplet number resulting from the subjective choice of activation scheme could potentially influence the magnitude of the indirect effect diagnosed from the model incorporating it.





## 1 Introduction

Ambient aerosol play a critical role in the climate system by interacting with radiation through several different mechanisms. Depending on its composition, aerosol can directly scatter or absorb incoming solar radiation, leading to a direct radiative effect and rapid changes in the energy budgets of the surface and atmosphere. Additionally, aerosol mediate the production of clouds

by providing surface area on which water vapor may condense to form droplets. Through this second pathway, changes in the aerosol population perturb the radiative properties of clouds by altering their microstructure and lifecycle, thereby impacting the planetary radiative budget. Despite decades of focused research by the scientific community, the radiative forcing produced through this second pathway, known as aerosol-cloud interactions, remains one of the largest uncertainties in understanding contemporary and future climate change on both regional and global scales (Boucher et al., 2013).

Contemporary Earth System Models are a valuable tool for assessing this uncertainty. Compared to their predecessors, these models incorporate aerosol microphysics schemes which represent the global variation in particle size distributions and aerosol composition. These aerosol size distributions can then be used to predict cloud droplet number concentrations. As a result, changes in the aerosol size distribution due to anthropogenic emissions can impact cloud optical properties and produce an aerosol-climate indirect effect. The interactions between aerosol particles, water vapor, and cloud droplets are often described

using the conceptual model of a possibly-entraining, adiabatic cloud parcel (e.g. Feingold and Heymsfield, 1992; Nenes et al., 2001; Ervens et al., 2005; Topping et al., 2013). This parcel model framework provides a convenient means for both direct numerical and laboratory simulation of aerosol-cloud interactions, and has been used to establish the physical basis through which change sin the available aerosol perturb cloud radiative processes (Feingold et al., 2001).

However, it is not practical to directly include parcel model calculations in global models due to their coarse grid scales.

Adiabatic parcel theory describes a process which occurs on a spatial scale of tens of meters, over the course of a few seconds—scales much finer than those resolved in many global model simulations. Some efforts have sought to incorporate fine-scale information about aerosol-cloud interactions by embedding higher-resolution models within each grid cell of a global model (for example, the "multi-scale modeling framework" of Wang et al., 2011). However, while this approach has improved the representation of cloud processes in global models, it still does not resolve the scales of motion associated with parcel

theory. To bridge this gap in spatial scales, global models employ activation parameterizations which predict cloud droplet number concentrations using information about the subgrid-scale variability in meteorology and aerosol size distribution.

Twomey (1959) first developed a scheme for placing an upper bound on this number of droplets derived from geometrical arguments applied to parcel theory. In subsequent decades, these arguments were used in conjunction with aircraft measurements to broadly characterize typical cloud droplet numbers in different regimes. For instance, Boucher and Lohmann (1995) devel-

oped power-law relationships between droplet number and sulfate mass concentration for separate continental and maritime regimes; similarly, Jones et al. (2001) developed a global parameterization for droplet number based on total aerosol number concentration using a survival function. While appropriate for earlier climate models which did not seek to resolve the complexity of the ambient aerosol, contemporary coupled aerosol-climate and aerosol-cloud-resolving models include more details about aerosol properties which can be used to predict cloud droplet number through explicit activation calculations. Broadly



speaking, efforts to improve these activation calculations have proceeded down two avenues: caching of pre-computed, detailed parcel model calculations; and extensions to Twomey's geometrical arguments using physically-based approximations.

The simplest technique applied to parameterizing aerosol activation is to pre-compute, using a detailed parcel model, a set of aerosol and meteorological conditions sampled over some pre-determined parameter space (Saleeby and Cotton, 2004; Ward et al., 2010). These samples can then be used to construct a look-up table, using some form of interpolation to compute activation results for points within the domain of the initial parameter space. However, changing or increasing the number of parameters—such as including additional aerosol modes or a composition dimension—in the look-up table rapidly increases its computational cost, since it is costly to both store and access the required information in memory during run-time.

Alternatively, one can extend Twomey's analytical approach to bounding the maximum supersaturation achieved in the constant-speed, adiabatic ascent of a parcel (e.g. Cohard et al., 1998; Cohard J-M. and Pinty, 2000). Several techniques have been employed to account for the sensitivity of the maximum supersaturation achieved by such a parcel to variations in the initial dry aerosol particle size distribution (Khvorostyanov and Curry, 2006; Shipway and Abel, 2010; Shipway, 2015). These techniques yield closed-form expressions which relate the parameters describing the initial aerosol size distribution and updraft speed to the supersaturation maximum, and can be used to predict droplet number concentrations for a given scenario.

Such studies have led to a number of explicit activation schemes, each applying a different computational and analytical approach but fundamentally relying on the same set of physical approximations (see Ghan et al., 2011, for a more complete review). Some of these schemes have been extended to account for additional physical or chemical processes which can influence activation, such as the presence of organic surfactants on the surface of droplets which tends to reduce surface tension (Abdul-Razzak and Ghan, 2004), or to account for entrainment in the ascending parcel (Barahona and Nenes, 2007). These schemes generally rely on iterative calculations to settle on an estimate of how many aerosol particles activate into cloud droplets (e.g. Shipway and Abel, 2010; Nenes and Seinfeld, 2003; Ming et al., 2006), although some employ pseudo-analytical solutions to avoid this process (Abdul-Razzak et al., 1998).

Since they offer generalized formulations independent of the representation of the aerosol particle size distribution provided to them, physically-based schemes have been the preferred tool for coupling with global aerosol-climate models. But, they are not without their shortcomings. Simplifying assumptions used to construct physically-based schemes, such as particles' equilibrium growth in response to changes in the ambient relative humidity, neglect kinetic limitations on growth and also lead to an underestimate in cloud droplet number in both polluted conditions and ones with weak updraft speeds (Nenes et al., 2001). Some schemes tend to under-estimate the number of droplets nucleated in the presence of multiple, competing aerosol modes, owing to their representation of water vapor uptake by particles (Simpson et al., 2014). Although Gantt et al. (2014) showed that using a variety of activation schemes in a modern aerosol-climate model with a complex aerosol particle size and composition distribution can lead to a global increase of estimated cloud droplet number by 155%, many evaluations of activation scheme performance have focused on the same set of relatively simple aerosol particle size distributions. For instance, Abdul-Razzak (2002) used the Whitby (1978) aerosol particle size distributions but with only two aerosol composition scenarios, varying the insoluble mass fraction in the coarse aerosol mode between either 0% or 90%. Nenes and Seinfeld (2003) and Fountoukis and Nenes (2005) used the same aerosol particle size distributions, but assumed just one additional aerosol composition scenario,



exploring the impact of aerosol which are composed of 20% (by mass) of an organic which displays surface-active behavior; Ming et al. (2006) employed an identical set of evaluation simulations. Shipway and Abel (2010) restricted the analysis of their scheme's performance with multi-modal aerosol particle size distributions to an idealized bi-modal distribution with non-varying, homogeneous particle composition. Ghan et al. (2011) summarized the performance of all of these simulations with

the same set of tri-modal aerosol particle size distributions, but extended their analysis to evaluate droplet number simulated by two schemes (Abdul-Razzak and Ghan, 2000; Fountoukis and Nenes, 2005) when employed in a global model. Meskhidze (2005) used in situ data collected from two different aircraft observation campaigns to evaluate the sectional formulations of the Nenes and Seinfeld (2003) and Fountoukis and Nenes (2005) schemes; these evaluations were extended by Fountoukis et al. (2007), which analyzed the modal formulations of the same schemes, and Morales et al. (2011) which showed that accounting

for entrainment reduced the over-prediction of droplet number in stratiform clouds by the activation schemes.

However, look-up table approaches do not necessarily solve these shortcomings. Detailed parcel models can more easily accommodate additional physics and chemistry which influences aerosol activation, but it would not be practical to incorporate this information into a look-up table. However, in previous work, Rothenberg and Wang (2016) developed a framework for producing an emulator of a detailed parcel model, which could be used to extend the benefits of look-up tables to high-

dimensional parameter spaces. In their example, which focused on the activation of a single lognormal aerosol mode, the meta-models produced by such a framework had lower error statistics on average than existing activation schemes when compared to a benchmark parcel model.

In this work, we extend the methodology developed in Rothenberg and Wang (2016) to develop an emulator suitable for inclusion in a modern, coupled aerosol-climate model. Furthermore, we assess the performance of the emulator against existing

activation schemes across a large input parameter space and a subset reproducing the tendencies of the aerosol-climate model. By reproducing the original, detailed parcel model on which it is trained, such an emulator could reduce biases in estimates of cloud droplet number concentration in cloud regimes characterized by either high pollution or relatively weak forcing and ascent. This could in turn lead to improved estimates of the aerosol indirect effect from global models.

This manuscript is organized to reflect the exploratory process used to develop the activation emulator presented here, with

the hope that clearly delineating the steps involved will encourage other groups to pursue similar lines of work. Section 2 provides background on the parent aerosol-climate model for which our activation emulator was derived. Section 3 details the construction of the emulators and the tools used to produce them. Two different evaluations of the emulators are presented in Section 4. Finally, in Sect. 5 we motivate future projects using these emulators and, more broadly, this approach to building parameterizations.

## 2   Activation Parameter Space

The parent aerosol-climate model for which we seek to derive a aerosol activation emulator is the MultiMode, two-Moment, Mixing-state resolving Model of Aerosols for Research of Climate (MARC; version 1.0.1 here) (Kim et al., 2008, 2014). MARC extends the NCAR Community Earth System Model (CESM; version 1.2.2 here), which is a fully-coupled global



climate model with sub-components for simulating climate processes in the land, ocean, atmosphere, and sea ice domains. In particular, MARC replaces the default modal aerosol treatment (Liu et al., 2012) in the atmosphere component of the CESM, the Community Atmosphere Model (CAM5; version 5.3 here), with a scheme which simultaneously resolves both an external mixture of different aerosol species and internal mixtures between others (Wilson et al., 2001). In this sense, MARC refers to

a configuration of the CESM with the CAM5 atmosphere component and the alternative aerosol formulation.

The aerosol population within MARC is comprised of a tri-modal sulfate distribution (nucleation ["NUC"], aitken ["AIT"], and accumulation ["ACC"] modes), discrete modes for pure black carbon ("BC") and a generic organic carbon species ("OC"), and two "mixed" modes, one of each for mixed sulfate-black-carbon ("MBS") and sulfate-organic-carbon particles ("MOS"). The ratio of the masses within each mixed species evolves over time, changing the optical and chemical properties of those

particles. For each of these seven modes, MARC predicts total particle mass ($M$) and number concentrations ($N$) for a corresponding lognormal size distribution with a prescribed width (geometric standard deviation; $\sigma_g$). Additionally, both sea salt ("SSLTn") and dust ("DSTn") particle size distributions are described within MARC by a 4-bin, single-moment scheme with fixed particle sizes. For these single moment modes, MARC predicts a total number concentration for each bin and then diagnoses a total mass in order to simulate a lognormal mode with prescribed geometric mean radius (which is narrower for

super-micron dust and sea-salt). Each mode has a prescribed particle hygroscopicity follow $\kappa$-Köhler theory (Petters and Kreidenweis, 2007) except for the MOS mode, which has a composition-dependent $\kappa$ computed assuming that the carbon and sulfate in the particle forms a simple internal mixture. Unlike the MOS mode, the MBS mode assumes a core-shell structure with sulfate coating a black carbon nucleus, and has a fixed hygroscopicity corresponding to that of sulfate, $\kappa = 0.507$. The sea salt modes are assumed to be comprised of NaCl with $\kappa = 1.16$, and the dust modes are assumed to be a mixture of

minerals with a hygroscopicity of $\kappa = 0.14$ (Scanza et al., 2014). The organic and black carbon modes are assumed to be non-hygroscopic and not significant players in aerosol activation. These assumptions about the aerosol size distribution simulated by MARC are summarized in Table 1.

The aerosol size distributions predicted by MARC interact with both radiation and cloud microphysics. With respect to the latter, MARC adopts the two-moment, stratiform cloud microphysics scheme found by default in CAM5 (Morrison and

Gettelman, 2008) which provides an explicit source of cloud droplet formation via aerosol activation from aerosol. This interaction is facilitated by means of a physical parameterization which takes as input both the physical and chemical properties of the ambient aerosol as well as limited information about meteorology—in particular, the distribution of subgrid-scale vertical velocities. In MARC, a single subgrid-scale characteristic updraft velocity ($V$) diagnosed from the grid cell turbulent kinetic energy (TKE) provided by the moist turbulence scheme (Park and Bretherton, 2009) and assumed to be isotropic is used to

estimate droplet nucleation following Ghan et al. (1997) and Lohmann et al. (1999), such that

$$V = \overline{V} + \sqrt{\frac{2}{3}\text{TKE}}$$

where $\overline{V}$ is the large-scale resolved updraft velocity. Furthermore, we limit $V > 0.2\,\text{m}\,\text{s}^{-1}$ because the processes driving turbulence are not represented well in MARC, particularly those driven by cloud-top radiative cooling above the planetary





boundary layer (Ghan et al., 1997). Morales and Nenes (2010) and West et al. (2014) have explored how using different characteristic updraft velocities to represent subgrid-scale variability can influence simulated aerosol-cloud interactions; in particular, West et al. (2014) showed that using a similar TKE-based parameterization produced more realistic spatial and temporal variability in $V$, but tends to produce an unrealistically high frequency of its minimum-permissible value. We further

assume that activation only occurs in non-entraining, adiabatic updrafts which carry air up and into the base of stratiform clouds.

## 2.1 MARC Aerosol and Meteorology Parameters

The set of size distribution parameters describing each aerosol mode in MARC and the number of meteorological factors influencing droplet nucleation in MARC is large, and each parameter can vary over several orders of magnitude across the

globe in even a single timestep. To assess this parameter space, we sample instantaneous snapshots of the 3D aerosol and meteorology fields from a MARC simulation run with present-day aerosol and precursor gas emissions. In total, 70 timesteps were sampled covering the complete seasonal and diurnal cycle at each model grid cell.

The variability in sub-grid scale vertical velocity as a function of continental versus maritime grid cells across all time samples in this output is summarized in Fig. 1. In both regimes, the mode updraft speed falls at the lower bound of $0.2\,\mathrm{m\,s^{-1}}$, and

occurs about 50% of the time. These velocities rarely exceed $1\,\mathrm{m\,s^{-1}}$ - 10% and 1% of the time over land and ocean, respectively. On average, land velocities are slightly larger ($0.41\,\mathrm{m\,s^{-1}}$ vs $0.32\,\mathrm{m\,s^{-1}}$), but have higher variance. The distribution of vertical velocities in both regimes has a long positive tail, maxing out between $3\,\mathrm{m\,s^{-1}}$ to $4\,\mathrm{m\,s^{-1}}$ and never approaching the artificial cap imposed by MARC.

The different particle size distributions in MARC are influenced by both different emissions sources and acted upon by

different physical processes. This leads to a great deal of spatial heterogeneity in the size distribution parameters. One aspect of this heterogeneity is depicted in Fig. 2, which shows distributions of the total number concentration of four of the modes aggregated into latitude and height bins. In general, number concentration for each mode decreases with height, as expected since (with the exception of the pure sulfate modes) all the modes have strong sources near the surface in the model. Natural aerosols (dust and sea salt) are generally much less abundant than anthropogenic ones. Furthermore, there are generally more

aerosol by number in the Northern Hemisphere than in the Southern Hemisphere to the preponderance of anthropogenic emissions sources.

Many of the distributions featured in Fig. 2 have long tails extending towards very low number concentrations. These very low values can be problematic for activation parameterizations, especially those constructed from statistical methods or sampling, such as a lookup table, as they necessitate many saved interpolation points. However, aerosol activation produces

at most one droplet per aerosol, so sensible lower bounds can be imposed to create a minimum threshold below which little activation is assumed to occur. Furthermore, particle size is a critical factor in assessing aerosol activation; larger particles have a much lower barrier to activation following Köhler theory (Seinfeld and Pandis, 2006). To simplify the assessment of how activation is influenced by the simulated aerosol size distributions in MARC, we diagnose $\mu_g$ from the prognostic moments output by MARC, and study it in lieu of the total mass concentration ($M$).



Although there is a great deal of variability in the number concentrations simulated by MARC, for all but the nucleation mode sulfate and coarse dust and sea salt modes, those values are most often drawn from a range of just a few orders of magnitude. In general, the number burden in each mode approximately scales with the total aerosol burden. Put another way, the number concentrations in the main sulfate and carbonaceous modes tend to correlate strongly with one another, as does each mode's corresponding mean size. In contrast, the number of sea salt particles tends to be much greater in the remote maritime environment where there are fewer particles overall (by number). The overall range of geometric mean mode particle sizes tends to be smaller than the range in number concentrations for each corresponding mode.

## 2.2 Reducing the Parameter Space

In total, MARC simulates 15 modes - seven double-moment and eight single-moment. As a result, we require 22 parameters to completely describe the aerosol size distribution. Two parameters are needed to close the description of its composition; the hygroscopicity and density of the MOS mode evolves in response to its relative mixture of carbon and sulfate, and the MBS mode accrues sulfate mass through aging, which also impacts its particle density. Finally, the ambient temperature and pressure, as well as the vertical velocity of the updraft in which activation is occurring are meteorological parameters which can influence the droplet number nucleated. This yields a total activation parameter space with 27 independent dimensions.

The emulation method used by Rothenberg and Wang (2016) is designed to work with an arbitrary number of input parameters. However, in order to focus our analysis on those parameters most influential on predictions of droplet nucleation, we restrict the input parameter space by eliminating parameters which exert little or no influence on the activation process. For instance, the pure carbonaceous modes (OC and BC) are hydrophobic and not sources of CCN, so we neglect them in the activation calculation. The nucleation mode sulfate typically has much fewer and much smaller particles than the Aitken mode and accumulation modes present in a grid cell. Furthermore, where nucleation mode particles are abundant in number, the other sulfate modes generally are, too. Thus, we also assume that the nucleation mode is not a source of CCN, and exclude it from our activation calculations. Additionally, the mixed black-carbon-sulfate mode (MBS) is assumed to be composed of particles with a carbon core and sulfate shell; we further assume that the entire surface of these particles are coated, effectively rendering the MBS particle hygroscopicity to be equal to that of sulfate. These assumptions effectively reduce the number of parameters we must consider by seven, bringing the initial parameters to a set of 17 aerosol ones and 3 meteorological ones.

## 2.3 Iterative Activation Calculations

To further reduce this number, we assess the relative importance of each individual aerosol mode and its influence on activation dynamics. This helps to identify a subset of aerosol modes to use as predictors in our emulator, avoiding the need to include all 17 potential aerosol parameters. We accomplish this with an ensemble of iterative activation calculations using a detailed reference parcel model (Rothenberg and Wang, 2016), drawn from a sample of aerosol size distributions simulated by MARC. This strategy effectively employs a "greedy" algorithm to sort the set of available aerosol modes, ranking their influence on activation by their cumulative depression on the supersaturation maximum achieved for a given parcel ascent.



Given a set of $n$ aerosol modes, the iterative calculations provide a sorted order for the modes, indicating their relative contribution to activation dynamics. Specifically, for each $n$ modes, we pick a test mode and run a parcel model simulation to compute the minimum supersaturation maximum ($S_{max}$) achieved in an updraft in which that mode is embedded. The mode which produces the minimum $S_{max}$ is said to be the "dominant" mode, and we record its size distribution. We then remove that aerosol from the original set of $n$ modes. At this point, we re-visit each of the $n-1$ remaining modes, and run parcel simulations in which the first "dominant" mode is present along with one additional mode. Again, we record the minimum $S_{max}$ and remove the contributing mode from the original set, adding it to the set of dominant modes. The end result of $n-1$ iterations is a complete sorting of the modes, based on their contribution to reducing $S_{max}$ in the parcel model simulations.

Figure 3 illustrates this iterative process for an example marine aerosol distribution following Whitby (1978). In this example, all particles are assumed to be pure sulfate. The vast majority of particles exist in the smaller, nucleation mode, but the coarse mode dominates the mass distribution. Only when inspecting the surface area distribution do all three modes even become apparent. With respect to activation, the number of aerosol particles is critically important, because droplets must form from individual particles. Thus, it would be reasonable to assume that the nucleation mode particles would "dominate" activation in this case - or that small changes in the burden of these particles could have large consequences on how many cloud droplets will form.

However, that doesn't happen. On the right-hand panel of Fig. 3 are plotted traces of the supersaturation achieved in a parcel with the indicated aerosol population, as a function of height above the parcel's initial altitude. In the first iteration, the parcel achieved minimum supersaturation maxima of 1.2%, 1.4%, and 1.3% when just the accumulation, coarse, and nucleation modes are present, respectively. These are much higher supersaturations than the Köhler-theory critical supersaturation for the geometric mean particle size in both the accumulation and coarse modes (0.3% and 0.1%, respectively), and a large fraction of the particles in those two modes activate. However, the number concentration of particles in those two modes is very small (60 and 3.1 particles $cm^{-3}$), and it takes time before the latent heat release due to condensation balances the cooling in the ascending parcels. Although the nucleation mode particles are much smaller and fewer of them activate, their number concentration is much higher ($340\,cm^{-3}$) and the total liquid water condensed on them is similar through the ascending parcel's trajectory, hence the similar values for $S_{max}$ in all three cases.

In the second iteration—with the accumulation mode omnipresent—the coarse and nucleation modes only reduce $S_{max}$ to 0.96% and 1.08%, respectively. The final iteration, which includes all three modes, produces an $S_{max}$ of 0.92% with an ordering of (accumulation, coarse, nucleation). These reductions in $S_{max}$ are due to the higher total particle number concentration and surface area available for condensation which leads to a more rapid balancing between the parcel's adiabatic cooling and warming due to latent heat release.

We apply this algorithm to a sample of 50,000 aerosol size distributions and meteorological parameters taken from our reference, present day MARC simulation. For each parameter set, we calculate the first four dominant modes, and record the supersaturation maxima produced by each successive combination. Using these supersaturation maxima, we diagnose the number of activated aerosol across the total aerosol population, including modes which are not present in the parcel model





simulation at a given iteration. Additionally, for each parameter set we perform a reference parcel model calculation where all aerosol modes are included, for comparison with the iterative calculations.

Overall, the accumulation mode sulfate (ACC) is the dominant mode in 96.5% of the sample cases. Infrequently, the mixed-sulfate-carbon modes and smallest dust mode are the dominant mode, accounting for all of the remaining cases. When ACC
dominates the activation dynamics, either the mixed-modes (MBS and MOS) or smallest sea salt mode is the second-most dominant one (in 10.3%, 36.2%, and 52.8% of cases, respectively). In fact, this ordering is so common that in 85% of cases, three of ACC, MOS, MBS, or SSLT01 comprise the top three dominant modes.

Figure 4 illustrates the potential error in calculating both $S_{\max}$ and $N_{\mathrm{act}}$ for each iteration of the activation calculations relative to using the full aerosol population, aggregated by which mode was the first-most dominant one. In all cases, using a
subset of the modes tends to over-predict the droplet number activation as a consequence of predicting a high value for $S_{\max}$. This is consistent with the physics of the activation problem; the presence of more aerosol surface area on which condensation can occur tends to produce a greater source of latent heat release to counter-balance adiabatic cooling in the ascending parcel, suppressing the development of a higher $S_{\max}$. But as Fig. 4 shows, this over-prediction decreases rapidly as modes are included in the calculation. Part of this decrease is related to the fact that adding modes in each iteration captures a higher fraction of the
total aerosol number; on average, the first dominant mode contains $70\% \pm 27\%$ of the total aerosol number, which increases to $80\% \pm 17\%$ and $89\% \pm 13\%$ after adding the second and third modes. Following this increase in fraction of the total aerosol number included by the dominant mode set in each iteration is a decrease in the absolute error in $N_{\mathrm{act}}$ relative to the full aerosol population, with an average of less than $1\,\mathrm{cm}^{-3}$ and max of $57\,\mathrm{cm}^{-3}$ by the third iteration.

With respect to the goal of reducing the aerosol parameter space necessary for assessing aerosol activation, what's more
important in these calculations than the frequently dominant modes, though, is the the absence of several modes altogether. In particular, the Aitken mode sulfate is never one of the first three dominant modes; beyond the modes depicted in Fig. 4, the only other modes in that set are the larger dust and sea salt modes. But in nearly all the cases sampled here, those modes have number concentrations that decrease as the bin's particle size increases. Also, the smallest dust and sea salt bins tend not to feature very high number concentrations in the first place. Even though the larger bins contribute particles which will
almost always activate, owing to their size and composition, the particles in them simply aren't numerous enough to impact the activation calculations.

## 3   Emulator Development

We now seek to construct an emulator of a detailed adiabatic cloud parcel model capable of resolving aerosol activation within an ascending, constant-speed updraft. The following sections briefly describe the chosen cloud parcel model and emulation
technique, polynomial chaos expansion. For more details on both techniques and their application, we refer the reader to Rothenberg and Wang (2016), which derives activation emulators for a simplified, single lognormal aerosol mode.



## 3.1 Parcel Model

Adiabatic cloud parcel models are a standard modeling tool for detailed assessments of aerosol activation and other studies focused on the composition of atmospheric particulates (Seinfeld and Pandis, 2006). In such a model, a constant-speed up-draft drives adiabatic cooling in a closed, zero-dimensional air parcel within which are any number or configuration of aerosol

particles. Initially prescribed a temperature, pressure, and water vapor content, the cooling parcel eventually develops a su-persaturation with respect to water vapor. In a sufficiently supersaturated environment, water vapor condenses on particulate surfaces. However, condensation releases latent heat, which counter-balances the parcel's adiabatic cooling. This balance can be expressed

$$\frac{dS}{dt} = \alpha(T,P)V - \gamma(T,P)\frac{dw_c}{dt} \tag{1}$$

where $V$ is the updraft speed, $\alpha(T,P) = (gM_wL/c_pRT^2) - (gM_a/RT)$ and $\gamma(T,P) = (PM_a/e_sM_w) + (M_wL^2/c_pRT^2)$ are functions weakly dependent on temperature and pressure (Leaitch et al., 1986), $M_w$ and $M_a$ are the molecular weights of water and air, $L$ is the latent heat of vaporization of water, $c_p$ is the specific heat of dry air at constant pressure, $R$ is the universal gas constant, $g$ is the acceleration due to gravity, $e_s$ is the saturation vapor pressure, and $w_c$ is the liquid cloud water mass mixing ratio (please refer to Appendix A of Rothenberg and Wang, 2016, for more details).

At some time $t$, the balance between heating due to latent heat release and cooling due to the parcel's adiabatic ascent will approximately balance such that $\frac{dS}{dt} = 0$ and a supersaturation maximum, $S_{\max}$, will occur. Thereafter, $S$ generally decreases, relaxing to some value close to unity as condensation drives droplet growth, quenching the ambient water vapor surplus. Beyond this point, the aerosol bifurcates into two populations: proto-cloud droplets which will continue to grow due to condensation and eventually collision and coalescence, and interstitial haze particles which may have become hydrated, but upon which

further condensation is not thermodynamically favorable.

In order to compute these effects, we simulate an arbitrary number of initial dry particle size distributions following a lognormal assumption:

$$n_N(r) = \frac{dN}{d\ln r} = \frac{N_t}{\sqrt{2\pi}\ln\sigma_g} \exp\left[-\frac{ln^2 r/\mu_g}{2\ln^2\sigma_g}\right] \tag{2}$$

Each aerosol mode is thus defined uniquely by three parameters $(N_t, \mu_g, \sigma_g)$ corresponding to the total aerosol number

concentration, the geometric mean particle radius, and the geometric standard deviation. The modes are each further discretized into a Lagrangian grid of 200 size bins, equally spaced over the logarithm of the particle radius such that each bin represents a different number of particles. The particles are then hydrated to an equilibrium size with respect to the initial relative humidity in the model; condensation acts to grow this size in response to the thermodynamic evolution of the parcel.

Furthermore, each bin is assigned a fixed hygroscopicity following $\kappa$-Köhler theory (Petters and Kreidenweis, 2007). By

tracking the dry particle size, a hygroscopicity, and the wet radius of each particle, aerosol activation can be directly assessed within the model.



### 3.2 Polynomial Chaos Expansion

We emulate the behavior of the detailed parcel model by applying the probabilistic collocation method (PCM; Tatang et al., 1997). PCM is a method of polynomial chaos expansion which seeks to construct a model response surface by mapping input parameters related to the initial conditions and behavior of a model to some response measured from the model. This process yields a computationally efficient yet accurate reproduction of the model.

The PCM is a non-intrusive technique which does not require modifications to an existing model in order to be applied. Instead, the PCM treats the original, full-complexity model as a black box and the chosen set of $M$ input parameters as independent, random variables, $\mathbf{X} = X_1, \ldots, X_M$, each with an associated probability density function. This PDF is used as a weighting function to derive a family of orthogonal polynomials which are used as the bases for the polynomial chaos expansion to be constructed, $\phi$. Using a finite number of these bases and choosing some model response, $R$, we write the polynomial chaos expansion as

$$R \approx \sum_{j=0}^{P} \alpha_j \phi_j(\mathbf{X}) \tag{3}$$

Such an expression has $N_t = P + 1 = (M + p)!/(M!p!)$ total terms, since a given chaos expansion of order $p$ will contain $p+1$ basis terms for each input parameter and combinations thereof. The coefficients $\alpha_j$ are computed by evaluating the original model at a set of particular set of sample points, recording the response of the model, and solving a regression problem. Those sample points are generated by taking the roots of the orthogonal polynomials associated with each of the input parameters and their random variables.

In order to compute the polynomial chaos expansions, we use the Design Analysis Kit for Optimization and Terascale Applications (DAKOTA; Adams et al., 2014), version 6.1. This software automates the process of generating input parameter sets, sampling the full-complexity model to be emulated, and constructing the polynomial chaos expansion. Furthermore, it provides many useful statistical properties of the sample dataset and the chaos expansions themselves.

### 3.3 Emulation of aerosol activation for MARC

We now apply the parcel model and chaos expansion technique in the previous sections to construct emulators of aerosol activation suitable for use in MARC. Following the analysis in Sect. 2, we identify a reduced-dimensionality input parameter space which covers the diverse set of aerosol and meteorology scenarios in which activation occurs in MARC, summarized by Table 2. Following the iterative calculations, we restrict the aerosol modes included in the activation calculation to just the accumulation mode sulfate and both mixed sulfate-organic carbon and sulfate-black carbon particles, as well as the two smallest dust modes and the smallest sea salt mode. To assess the importance of these less-abundant, coarse particles, we derive two emulators: a "main" scheme which includes just the ACC, MBS, and MOS modes, and a "gCCN" scheme which adds in the dust and sea salt modes. All the aerosol size distribution parameters are transformed using a logarithm, since they can





take on values that span several orders of magnitude. Additionally, we consider the hygroscopicity of the mixed sulfate-organic carbon mode, as well as the updraft speed and ambient temperature and pressure in our input parameter space.

For each of these parameters we construct uniform size distributions, which are uniquely defined by a set of low and high bounds. These bounds are also noted in Table 2, along with the percentile of the data they correspond to from our sampling

study of the parameter space for the aerosol parameters. For most of those parameters, the bounds cover upwards of 99% of the sampled parameter space. Notably, the lower cut-off boundaries for dust and sea salt number concentration occur at much higher percentiles; the number concentration of droplets nucleated in our calculations is relatively insensitive to changes in the coarse mode number concentration at very low values, though, so we opt to constrain the input parameter space to these modes to a more physically relevant range. The MOS hygroscopicity, temperature, pressure, and updraft velocity ranges cover

all plausible values that could be used in an online activation calculation within MARC.

We cast all of the input parameters as random variables with uniform probability density functions to emphasize that we care equally about computing activation with a parameter set drawn with any parameter values. However, in MARC, some combinations of parameters are extremely unlikely. For instance, sea salt has sources far removed spatially from the sources of black carbon; ergo, it is uncommon to see a high number concentration for sea salt as well as the mixed sulfate-black carbon

mode. The trade-off here potentially lies in emulator performance, because the PCM will attempt to train the emulator to perform well for input parameters that we've assigned equal likelihood to, but may actually be far less likely to occur. A major benefit from this trade-off is simplicity in matriculating the emulator for use in a GCM. Using multiple probability density functions for different input variables, while offering a tuning knob to increase the accuracy of the emulators, would require a larger set of orthogonal polynomial bases than just the Legendre polynomials used for the uniform distributions here.

These parameters are used to drive parcel model simulations where we record the logarithm of $S_{max}$ as the response variable. This value can then be used to diagnose the number concentration of droplets nucleated by assuming that any particles which experience their Köhler theory-predicted critical supersaturation. We note that although this does not resolve the issue of kinetic limitations on droplet growth and its potential to cause an under-prediction in droplet number (Nenes et al., 2001), unlike existing activation schemes, our emulator accounts for the feedback of these effects on $S_{max}$, so one avenue of its impact

is lessened. Furthermore, emulators predicting $S_{max}$ were much more accurate at reproducing parcel model behavior than those which directly predicted estimates of $N_{act}$ accounting for kinetic limitations on growth.

The end result of constructing the emulators is a function which maps $\log_{10}(S_{max})$ to a set of values from our input parameter space,

$$\log_{10}(S_{max}) = f(\log_{10} N_{ACC}, \log_{10} N_{MOS}, \log_{10} N_{MBS},$$
$$\log_{10} \mu_{ACC}, \log_{10} \mu_{MOS}, \log_{10} \mu_{MBS}, \kappa_{MOS}, \log_{10} V, P, T[, m$$
$$\log_{10} N_{DST01}, \log_{10} N_{DST02}, \log_{10} N_{SSLT01}]) \tag{4}$$



From a prediction of the $S_{\mathrm{max}}$ achieved in an ascending parcel with the conditions passed to the emulator, we can then diagnose aerosol activation by re-writing the lognormal size distribution for each mode as a function of critical supersaturation (Ghan et al., 2011) to yield an expression

$$N_{\mathrm{act}} = \sum_{i=1}^{n} \frac{N_{t,i}}{2} \left( 1 - \mathrm{erf}\left[ 2\ln\left(\frac{S_{m,i}}{S_{\mathrm{max}}}\right) / (3\sqrt{2}\ln\sigma_{g,i}) \right] \right) \tag{5}$$

where $S_{m,i}$ is the critical supersaturation for a particle of radius $\mu_{g,i}$ from mode $i$.

## 4   Evaluation of Emulators

We evaluate our emulators by applying them to both a synthetic sample of potential input parameters as well as real samples taken from a MARC simulation. In all of our comparisons, we study third and fourth order chaos expansions both excluding ("main") and including ("gCCN") the coarse dust and sea salt modes.

As a reference, we compute activation statistics for each sample from several different sources. First, we run the detailed parcel which the emulator aims to simulate. Second, as a further benchmark and comparison, we run two widely-used activation parameterizations from the literature. The first scheme, by Abdul-Razzak and Ghan (2000) (ARG) uses a pseudo-analytical solution to an integro-differential equation derived from the original adiabatic parcel model system. However, one part of the pseudo-analytical calculation involves a fit to parcel model calculations. The second parameterization, by Morales Betancourt

and Nenes (2014b) (MBN), applies an iterative scheme to partition the aerosol population into two subsets, and uses different limits on the underlying analytical formulas to derive a maximum supersaturation. Because it requires a sequence of iterations to run, the MBN scheme is more computationally expensive than the ARG scheme, but has the potential to include more detailed links between particle composition and condensation (Kumar et al., 2009) or entrainment into the parcel (Barahona and Nenes, 2007). Like the ARG scheme, though, one limiting case in the MBN scheme relies on a fit to parcel model simulations. In both

cases, those simulations involved models conceptually similar to the one emulated here.

### 4.1   Input Parameter Space Sampling

Using the parameter space defined in Table 2, $n = 10000$ sample parameter sets were drawn using maximin Latin Hypercube Sampling (LHS). This randomized sampling method helps to ensure that the full aerosol and meteorology parameter space is studied while assessing its performance.

Figure 5 compares the performance of each emulator and the two reference activation scheme against parcel model simulations using all of the LHS samples for the "main" aerosol parameter sets. In the simulations, higher updraft speeds (shaded) are nearly always associated with a much higher supersaturation maximum. For the emulators, accuracy tends to increase on average going from the 3rd order (Fig. 5-a) to the 4th order (Fig. 5-b) scheme, although there is slightly higher variance in the relative error compared to the parcel model at higher updraft speeds. With respect to the driving updraft speed, though, there

isn't a consistent mode of bias - on average, the relative error is very low. The same does not hold true for the two reference





schemes. The ARG scheme (Fig. 5-c) tends to predict both too-high and too-low supersaturation maxima at higher updraft speeds but is relatively well-calibrated at lower updraft speeds yielding a lower supersaturation maximum. On the other hand, the MBN scheme (Fig. 5-d) is generally more accurate and better-calibrated than either of the emulators or the ARG scheme, especially at higher updraft speeds - but tends to spuriously over-estimate $S_{\mathrm{max}}$ for weak updraft speeds.

Figure 6 takes the results depicted in Fig. 5 one step further by diagnosing droplet number concentrations nucleated from each $S_{\mathrm{max}}$. For all the schemes, there can be substantial differences between the parcel model and each parameterization. This is particularly the case in regimes which give rise to fewer overall droplet number concentrations, either due to a lower driving updraft velocity, or a lower total aerosol number available to activate. Surprisingly, the MBN scheme tends to consistently activate a higher number of droplets with respect to the parcel model, especially in situations which should have very few

droplets - below $10\,\mathrm{cm}^{-3}$. The ARG scheme does not have as consistent of a bias, but can both egregiously over-predict and under-predict droplet number, with these biases exaggerated at lower updraft speeds. By comparison, the emulators show much less overall bias. The mean error for the emulators follows that of $S_{\mathrm{max}}$ and is small, but there is variance which tends to impart a small low or high bias on its estimates.

Both of these sets of plots are repeated in Figs. 7 and 8, but for the "gCCN" experiment. Qualitatively, the results for all

parameterizations are very similar, with the same overall biases - especially for the ARG and MBN parameterizations. The emulators tend not to perform as well overall in the "gCCN" cases, although they are still the most highly-calibrated scheme and do not have the velocity-regime errors that the MBN scheme has. In both the "main" and "gCCN" parameter sets, the MBN scheme tends to more regularly predict too many cloud droplets, save for polluted regimes giving rise to $100\,\mathrm{cm}^{-3}$ droplets where that bias reverses and the scheme has a tendency to under-predict droplet number. Neither the emulators nor the ARG

show this same tendency in bias.

These differences in bias are most likely related to the choice of parcel model used in testing and building the ARG and MBN schemes; because each scheme relies on some empirical tuning to parcel model calculations, details in the implementation of each parcel model which influence its sensitivity should show up ensemble evaluations of each activation scheme. The "gCCN" case is more taxing to simulate with parcel models using a Lagrangian description of the particle size distribution, because

condensational growth is computed for each particle bin simultaneously. The stiffness ratio in this case will be extremely large, as the small particles in the main aerosol modes will grow much more slowly than those in the giant CCN modes. Although modern ODE solvers can automatically handle these scenarios, the subjective choice of which particular solver and how to discretize the giant CCN population (how many bins per mode) could greatly influence the sensitivity of $S_{\mathrm{max}}$ to changes in the model inputs and account for the differences observed here.

To better summarize the results in Figs. 5 to 8, summary statistics on the error of each scheme versus their corresponding parcel model calculations are shown in Table 3. In both sampling cases, all of the parameterizations show a strong linear correlation ($r^2$) between their predictions and the result of the parcel model. The emulators (PCM Order $p$) predict $S_{\mathrm{max}}$ with lower overall absolute and relative error, but with a much higher variance (not shown here). However, that lower error does not always translate into the emulators being the most accurate absolute predictors of $N_{\mathrm{act}}$. For the "gCCN" parameters, the ARG





scheme predicts $N_{\mathrm{act}}$ with a lower average mean relative error. In both parameter sets, the MBN scheme is the least accurate compared to the parcel model used in these sampling calculations.

## 4.2 MARC Aerosol Sampling

Although the sampling in the previous section fully exhausts the input parameter space over which aerosol activation may
need to be assessed, it undoubtedly samples from aerosol and meteorological conditions which may not be likely to occur in the real world. To better understand the performance and potential bias of the emulators developed here and the existing activation schemes, then, we also studied a sample of $n = 10000$ aerosol and meteorology parameter sets drawn directly from a MARC simulation. All of the schemes were evaluated again using these parameter sets and the detailed parcel model. This includes the "main" and "gCCN" emulators, which allows us to identify the importance of including the dust and sea salt
modes as predictors in the chaos expansions. The parameters in these sets occasionally include values outside the ranges defined in Table 2 and studied in the previous section. These cases are more frequently associated with very low total aerosol number concentration, especially over the ocean where anthropogenic aerosols are limited and natural aerosols - which have a lower overall number burden - dominate. Because the aerosol samples from oceanic and continental grid cells differ in this fundamental way, we break down the following analysis to reflect those differences. As in the previous sampling experiment,
summary statistics on the performance of each emulator, alongside the ARG and MBN schemes, are detailed in Table 4.

Qualitatively, all of the activation schemes perform similarly when evaluated against the MARC parameters as compared to the more generic sampling in the previous section. Figure 9 summarizes distributions of relative error in $N_{\mathrm{act}}$ over land and ocean for each scheme. Neither the ARG nor the MBN scheme show much difference in error for the two regimes, although on average, the MBN tends to under-predict $N_{\mathrm{act}}$. This under-prediction usually occurs in regimes with higher updraft speeds and
thus higher overall droplet number concentrations. In conditions with weaker updraft speeds, the MBN scheme instead tends to slightly over-predict $N_{\mathrm{act}}$. The ARG scheme is particularly well-calibrated in both regimes.

The emulators derived here do not fare as well as the physically-based parameterizations. Both 3rd order schemes tend to over-predict droplet number over oceans, and under-predict it over land, but with an extremely large variance extending to $\pm 100\%$. However, including the effects of giant CCN measurably improves the performance of the 3rd order emulators in
oceanic regimes. Increasing the order of the emulator also has a significant impact on their accuracy; the 4th order scheme which neglects giant CCN actually out-performs the ARG and MBN scheme on average, and shows little bias between land and ocean regimes, indicating good convergence with its parent parcel model. On the other hand, the gCCN scheme has not yet converged by including 4th order terms, even while its mean error statistics improve. Particularly troublesome is a secondary mode of extreme (over 100%) under-prediction of droplet number of oceanic regimes, but this metric is deceptive. Really,
what is occurring is that for very low total aerosol number concentrations - with particle number in the single-digits per cubic centimeter - the 4th order "gCCN" scheme tends to predict half as many droplets as parcel model calculations indicate should form. This typically occurs when one or more of the input size distribution parameters (in particular, the number concentration) for the natural aerosol dips below the minimum threshold where the emulator was trained. When the emulators encounter inputs greater (lower) than these thresholds, they hold them to the maximum (minimum) value in its training range This follows the



assumption that the bounds for each parameter cover the entire range over which activation is sensitive to changes in that input. Put another way, activation should be relatively insensitive to changes near the maximum or minimum values in the range for each parameter. With respect to number concentration, this must be the case; populations with fewer than $10^{-3}$ particles cm$^{-3}$ offer very little surface area for condensation, and simply cannot exert a strong influence over the developing supersaturation

in the parcel. That the 4th order "gCCN" emulator produces too high of sensitivity in this regime suggests that statistical over-fitting is occurring near the extremes of the input parameter space.

To contextualize these differences in $N_{act}$ bias over different geographical regimes, Fig. 10 re-maps the testing samples back to the original MARC grid. Here, the difference in regional biases becomes much clearer. Virtually everywhere, the MBN scheme is biased a little low, but there is no systematic difference in this bias between land or ocean, or by geographical areas.

The ARG scheme and the 4th order "main" scheme show a different pattern; the ocean-land contrast in bias is clearly visible in the northern hemisphere. Furthermore, the bias is typically positive over maritime regions, but negative over regions with anthropogenic aerosol influence. In particular, these regions include Europe and southeastern Asia - where aerosol distributions are dominated by anthropogenic sulfate and black carbon - and over north central Africa - where the aerosol is a mixture of both dust and organic carbon emissions from biomass burning. In the zonal average, the main_4 scheme is virtually identical

to the ARG scheme. However, both cases as well as with MBN, there are larger biases over the southern parts of the oceans, where the aerosol is predominantly comprised of sea salt and smaller sulfate particles produced indirectly through the emission of DMS.

Figure 10 also illustrates the poor performance of the gCCN_4 scheme, which under-predicts $N_{act}$ nearly everywhere, but especially so in the southern portions of the ocean basins. The consistent under-prediction in this region explains the bimodal

distribution over the ocean hinted at in Fig. 9. The gCCN_4 scheme does not perform too dissimilarly than the other schemes over regions with anthropogenic pollution or with mostly dust aerosol.

## 5  Discussion and Conclusions

In this work, we extended the meta-modeling technique of Rothenberg and Wang (2016) in order to apply it to assess aerosol activation of a complex, multi-modal aerosol mixture simulated by a modern aerosol-climate model. Simultaneously, we char-

acterize the performance of both our new emulators for aerosol activation and two widely-used schemes from the literature, focusing on that same high-dimensional, complex aerosol parameter space. To identify the most important factors impacting activation in that complex parameter space, we apply a physically-based approach to assess the sensitivity of activation statistics to the composition of the aerosol size distribution. Finally, we explore contrasts between aerosol and meteorology regimes over land and ocean, noting the potential for different biases in assessed cloud droplet number depending on the choice of

activation scheme used in a particular global modeling application.

In ensembles of iterative calculations using a large sample of aerosol size distributions from a coupled aerosol-climate model, we note that typically, a single mode tends to dominate activation or otherwise strongly predict the total number of droplets nucleated. This approach to understanding the sensitivities of activation dynamics on the underlying aerosol population is



distinct from previously-published approaches in the literature. For instance, Karydis et al. (2012) and Morales Betancourt and Nenes (2014a) apply an adjoint approach to derive the sensitivity of aerosol activation to perturbations in input parameters supplied to activation schemes. Detailed calculations using this approach yield a map of local sensitivities or gradients in the relationship between, for example, $N_{\mathrm{act}}$ and one input parameter while holding all others constant, and are thus difficult to

interpret. The iterative calculations performed here aim instead to address the global sensitivity of activation to configurations of an aerosol population.

It is somewhat surprising that the accumulation mode sulfate (ACC) successfully serves as such a strong proxy for the full aerosol population, even in the presence of giant CCN and a wide swath myriad updraft regimes. This result is likely model-dependent; the ACC mode in MARC is not only ubiquitous, but may be inadvertently (and subjectively) in a range

of mean particle sizes for which aerosol activation is especially sensitive. At the same time, the coarse dust and sea salt modes in MARC, on average, hold too small a number concentration to dramatically impact activation calculations save for remote maritime regions far removed from anthropogenic sources. However, the presence of sea salt as one of the modes most frequently ranked in the top three influencers of activation points to previous results which indicate the presence of giant CCN influence activation dynamics (e.g. Barahona et al., 2010).

The fact that a single mode can place such a strong constraint on aerosol activation is useful for attempts seeking to extend look-up table methods for building parameterizations. If two modes—an accumulation-size and a coarse-size—accurately predict aerosol activation, then one can constrain the look-up table to just a few key aerosol size distribution parameters. The inclusion of variable aerosol composition would still likely make employing a look-up table in a global model unwieldy, though, necessitating more sophisticated approaches. The emulation technique applied here is one such approach to tackling

this problem which appears to work very well.

When sampling against the full training parameter space employed here, our emulators perform capably. Neglecting the influence of the giant CCN modes, the emulators built average a mean relative error of less than 1% in predicting $\log_{10} S_{\mathrm{max}}$, which translates to an mean relative error of 9.2% and 8.9% in predicting $N_{\mathrm{act}}$. Including the giant CCN mode appears, at first, to dramatically increase the performance of the emulator, bringing those same metrics down to 0.3% and 6.9% for the 4th-order

scheme. Relative to the ARG and MBN schemes, the emulators are much more accurate on average when compared to our reference parcel model. However, we note that both the ARG and MBN schemes contain components which themselves are tuned to parcel models employed by their developers. Thus, we should not expect those schemes to perfectly match the parcel model results calculated here. Instead, we emphasize that the comparison of our emulators with the ARG and MBN scheme is motivated as part of a broader attempt to understand how the fundamental activation process initiates a chain of physics which

ultimately lead to the aerosol indirect effect on climate. Assessing the relative performance of activation schemes which, for all intents and purposes, perform extremely well at reproducing their own reference parcel models, is a critical step in establishing the parametric uncertainty in translating aerosol to droplet numbers and which underlies uncertainty in global model estimates of the indirect effect.

For this reason, we supplemented the evaluation of our emulators by using a second set of input parameter samples drawn

from aerosol fields simulated by an aerosol-climate model. In contrast with previous studies, we use instantaneous fields in





lieu of monthly or annual averages for our samples. Activation is inherently a fast process; because the microphysics schemes in aerosol-cloud models directly account for a tendency of new droplets formed via nucleation, the activation parameterization in any model will be called every time-step and for every grid-cell where clouds are occurring. Assessing activation schemes using temporally-averaged aerosol fields risks missing some combinations of input parameters and limiting the range of values

for which the scheme will need to accurately perform.

Most of the emulators and schemes tested here perform somewhat differently over land and ocean, owing to the presence (or lack thereof) of natural and anthropogenic aerosols in these different regimes. Unfortunately, when focusing on the narrower range of aerosol parameters present in MARC (in comparison with the larger parameter space on which the emulators were trained), the emulators which explicitly account for giant CCN perform poorly, especially over ocean regimes dominated by

sea salt. However, their counterpart performs nearly identical to the ARG scheme, showing a slight over-prediction of $N_{act}$ in maritime regimes and a slight under-prediction over continents. In the global average, the emulator agrees better with the detailed parcel model than the ARG scheme. By comparison, the MBN scheme, while prone to under-predicting $N_{act}$ in both regimes in these calculations, shows far less variance in its mis-prediction. This would suggest the MBN scheme actually performs extremely well - it is simply calibrated against a different baseline (in this case, a different parcel model). In particular,

the MBN scheme does not show a difference in relative error between ocean and land regions, suggesting it is appropriately sensitive to a large range of different aerosol populations.

The results presented here have important implications for global modeling studies seeking to quantify uncertain in the aerosol indirect effect on climate. While different activation schemes generally perform equally well when faced with idealized sets of input parameters (Ghan et al., 2011), their application in coupled aerosol-climate models may not be straightforward.

Relative to parcel model calculations, activation schemes can likely show biases in predicting cloud droplet number in different regions of the world owing to spatial heterogeneity in the underlying aerosol and meteorology parameter distributions. This, in turn, will lead to biases in cloud radiative forcing and diagnosis of the indirect effect.

Some literature has already implicated the role of activation schemes in divergent model estimates of the indirect effect. Ghan et al. (2011) performed a pair of GCM experiments using two different schemes and noted that between their simulations there

is a 10% difference in the global average droplet number concentration, which produces a $0.2\,\mathrm{W\,m^{-2}}$ difference in the indirect effect. Using a sequence of increasingly-complex activation schemes, Gantt et al. (2014) performed similar simulations with just a present-day emissions scenario, showing large regional differences in average cloud droplet number concentration and, as a result, up to a difference of $0.9\,\mathrm{W\,m^{-2}}$ in global average shortwave cloud forcing. This perturbation in forcing naturally follows from results such as those highlighted in this work; changes in the base cloud droplet number concentration simulated

in an aerosol-climate model have important consequences for the chain of cloud microphysical processes which ultimately give rise to the indirect effect. Those biases will necessarily be model-dependent, since the formulation of the basic activation diagnostic in each model is intertwined with regional and global variability in their simulated aerosol size distributions.

Future work should seek to systematically assess the differences in cloud microphysical processes and aerosol-cloud inter-actions arising from choice of activation schemes in aerosol-climate models. As this work illustrates, employing emulators of

detailed parcel model calculations including various chemical and physical effects on the activation process will aid with this





task, providing a way to quickly account for myriad facts which may be difficult or impossible to include in existing activation schemes or frameworks. However, this work must proceed in tandem with efforts to place strong constraints on the climatology and variability of cloud droplet number concentration across regions and meteorological regimes. The synthesis of these two lines of work may provide the necessary constraints to diagnose systematic biases in the representation of fundamental aerosol-

5  cloud interactions in global aerosol-climate models and thus reconcile the disagreement between model- and satellite-derived estimates of the indirect effect.

## Appendix A:  Code and Data Availability

A git repository archiving the scripts used to generate the chaos expansions can be found at https://github.mit.edu/darothen/marc_pcm_activation. For the convenience of the reader, an up-to-date commit (c71f8ca9bd4) has been included in the Supple-

10  mentary Materials. Dependencies of these scripts are recorded in the README file therein. The sampling datasets generated for analysis in this work are archived with DOI 10.5281/zenodo.60937. The source code and documentation for the pyrcel cloud parcel model are archived with DOI 10.5281/zenodo.46127, and can be accessed from http://github.com/darothen/pyrcel.

*Acknowledgements.*  The work in this study was supported by the National Science Foundation Graduate Research Fellowship Program under both NSF Grant 1122374 and NSF Grant AGS-1339264; the National Research Foundation of Singapore through the Singapore–MIT

15  Alliance for Research and Technology and the interdisciplinary research group of the Center for Environmental Sensing and Modeling; and the U.S. Department of Energy, Office of Science (DE-FG02-94ER61937). We thank Steve Ghan (PNNL) and Athanasios Nenes (Georgia Tech) for reference implementations of their activation parameterizations. We would also like to thank Topical Editor Graham Mann for comments that helped improve this manuscript.



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

**Figure 1.** Distributions of model-predicted instantaneous sub-grid scale vertical velocity for near-surface (below 700 mb) grid-cells broken down by land (red) and ocean (black) regimes.





**Figure 2.** Violinplots showing the distribution of the logarithm of the number concentration for total dust (DST), total sea salt (SSLT), accumulation mode (ACC) and mixed sulfate-black-carbon mode (MBS) aggregated by latitude (columns) and vertical level (rows). The width of each violinplot is scaled by the number of observations for each mode in a given aggregation. The inner boxplot on each figure shows the median and interquartile range for reference.



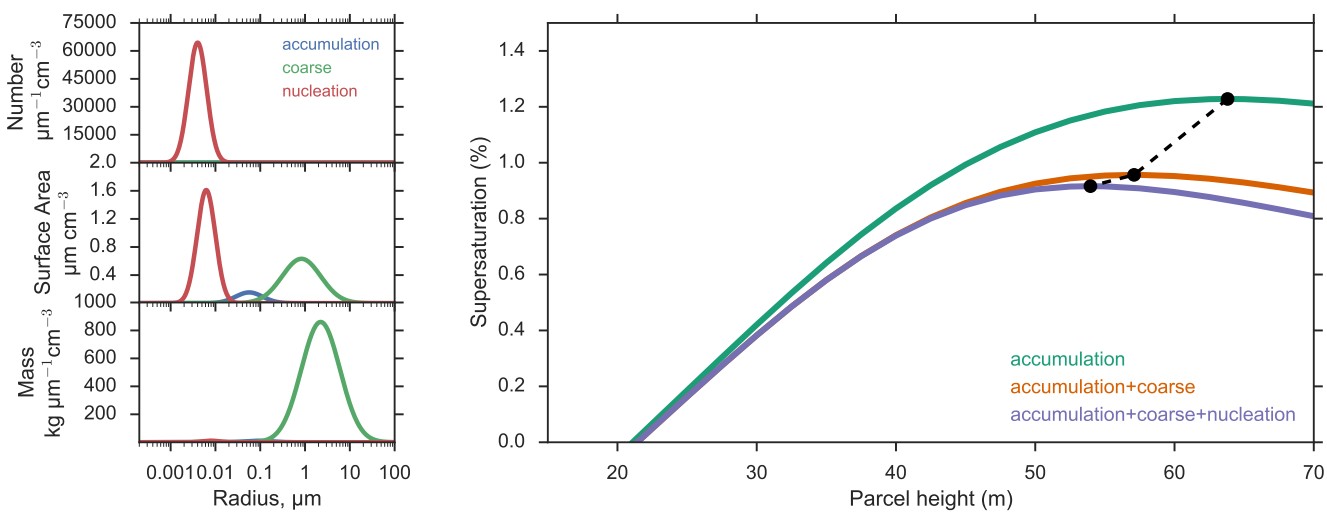

**Figure 3.** Illustration of particle number, surface area, and mass distributions (left) for typical marine size distribution from Whitby (1978), along with iterative activation calculations (right). In all calculations, we assume the aerosol are all sulfate particles with a hygroscopicity of $\kappa = 0.56$, and are activated in constant-speed updraft of $V = 0.5\,\mathrm{m\,s^{-1}}$

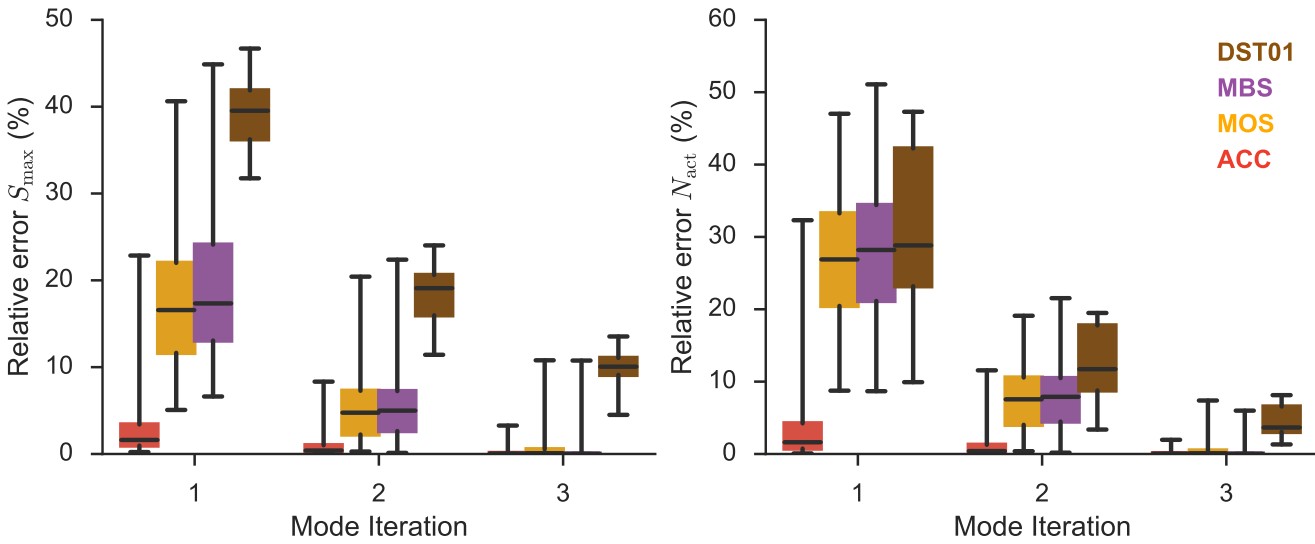

**Figure 4.** Relative errors in $S_{\mathrm{max}}$ (left) and $N_{\mathrm{act}}$ (right) in subsequent iterations of the iterative activation calculations. The coloring of each box indicates which mode was the first or dominant one. In each boxplot, box encompasses the interquartile range and the whiskers extend to the 1st and 99th percentiles in the corresponding sub-sample. Outliers beyond these percentiles are not plotted.





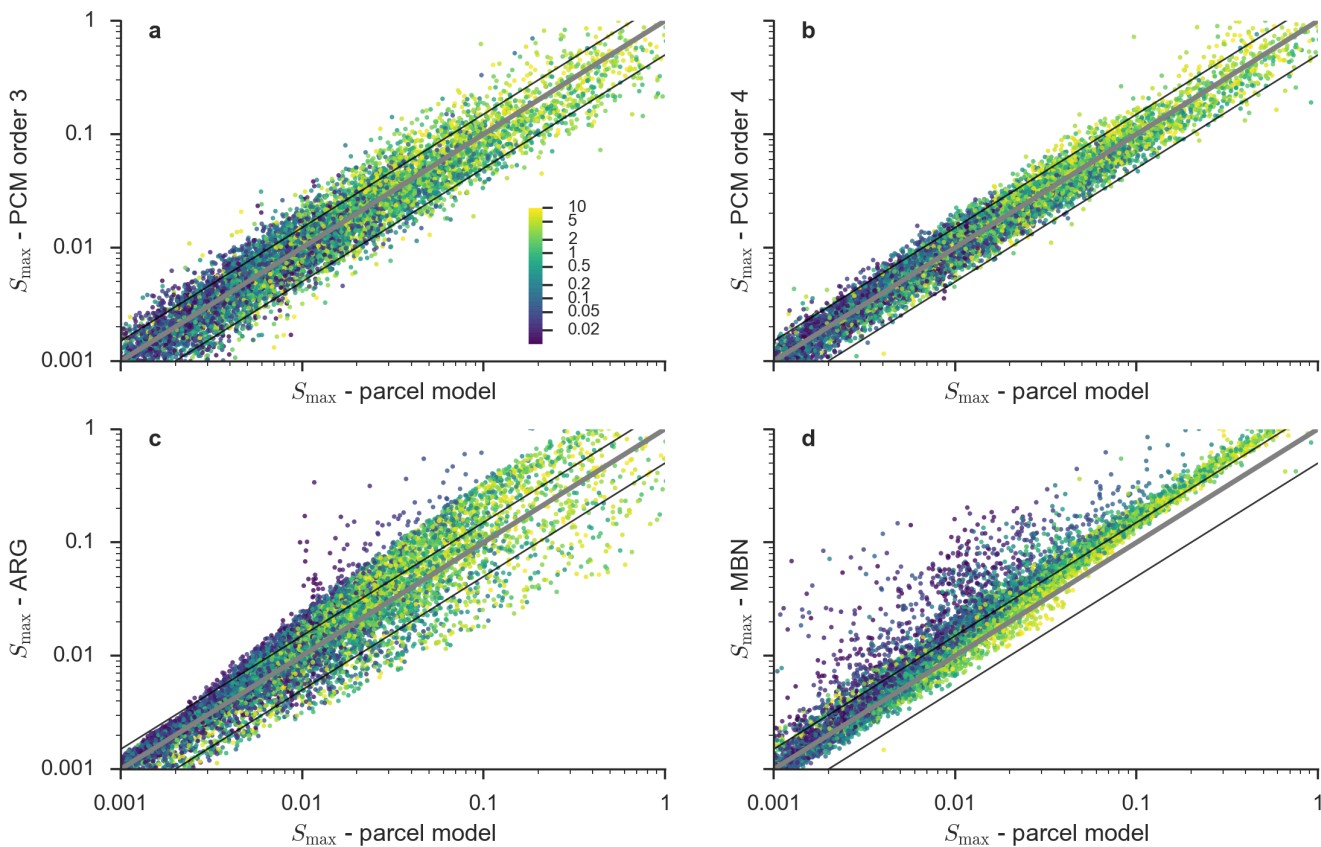

**Figure 5.** One-one plot comparing predicted supersaturation maxima between parcel model and activation parameterizations - 3rd-order emulator (a), 4th-order emulator (b), ARG (c) and MBN (d). The "main" aerosol parameter set (excluding the dust and sea salt as predictor modes) were utilized here. Glyphs are shaded to denote updraft velocity corresponding to each sample draw (in m/s), and are consistent for each panel. Solid black lines denote a factor-of-2 difference between predicted values from parcel model and corresponding parameterization evaluations



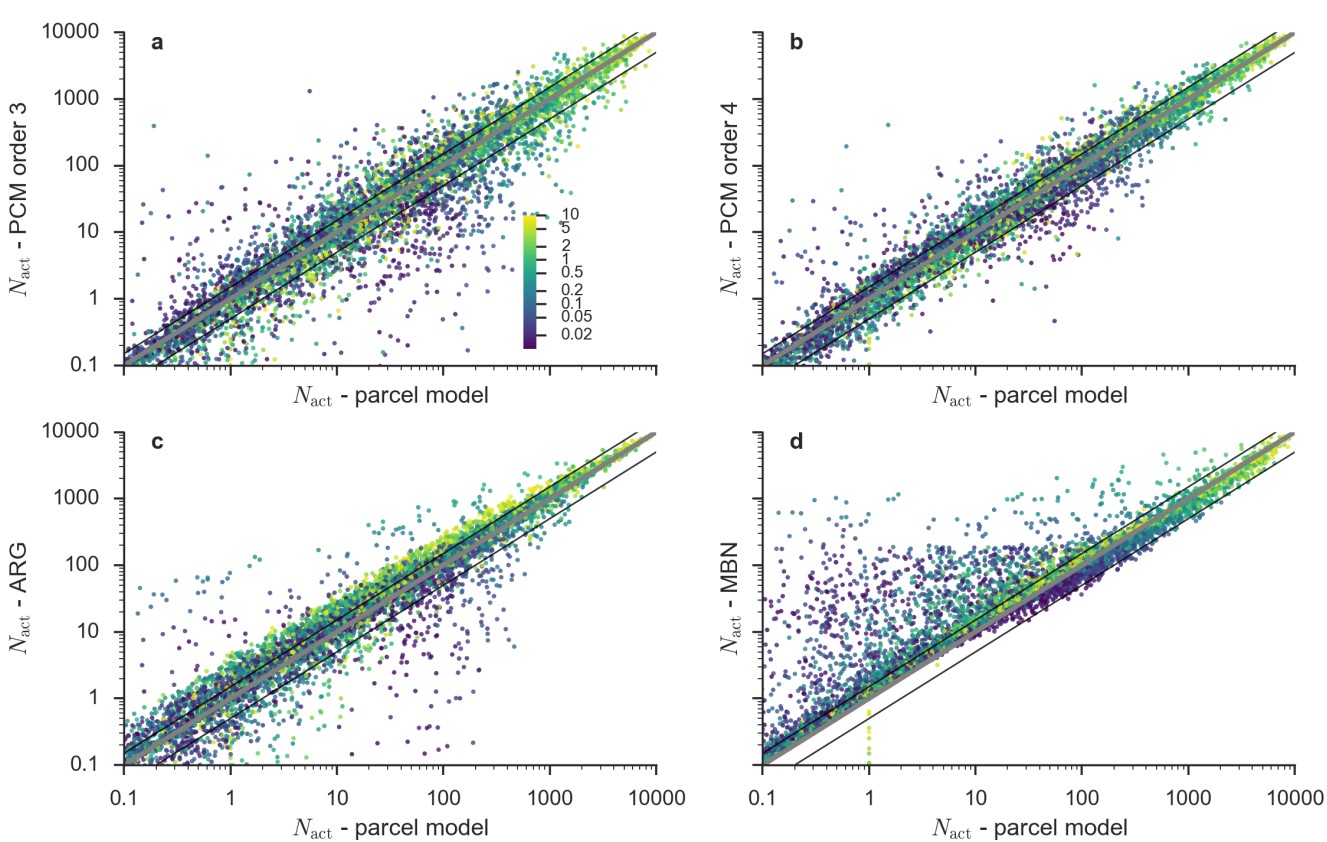

**Figure 6.** Like Figure 5, but plotting the predicted droplet number concentration nucleated for the aerosol "main" parameter set





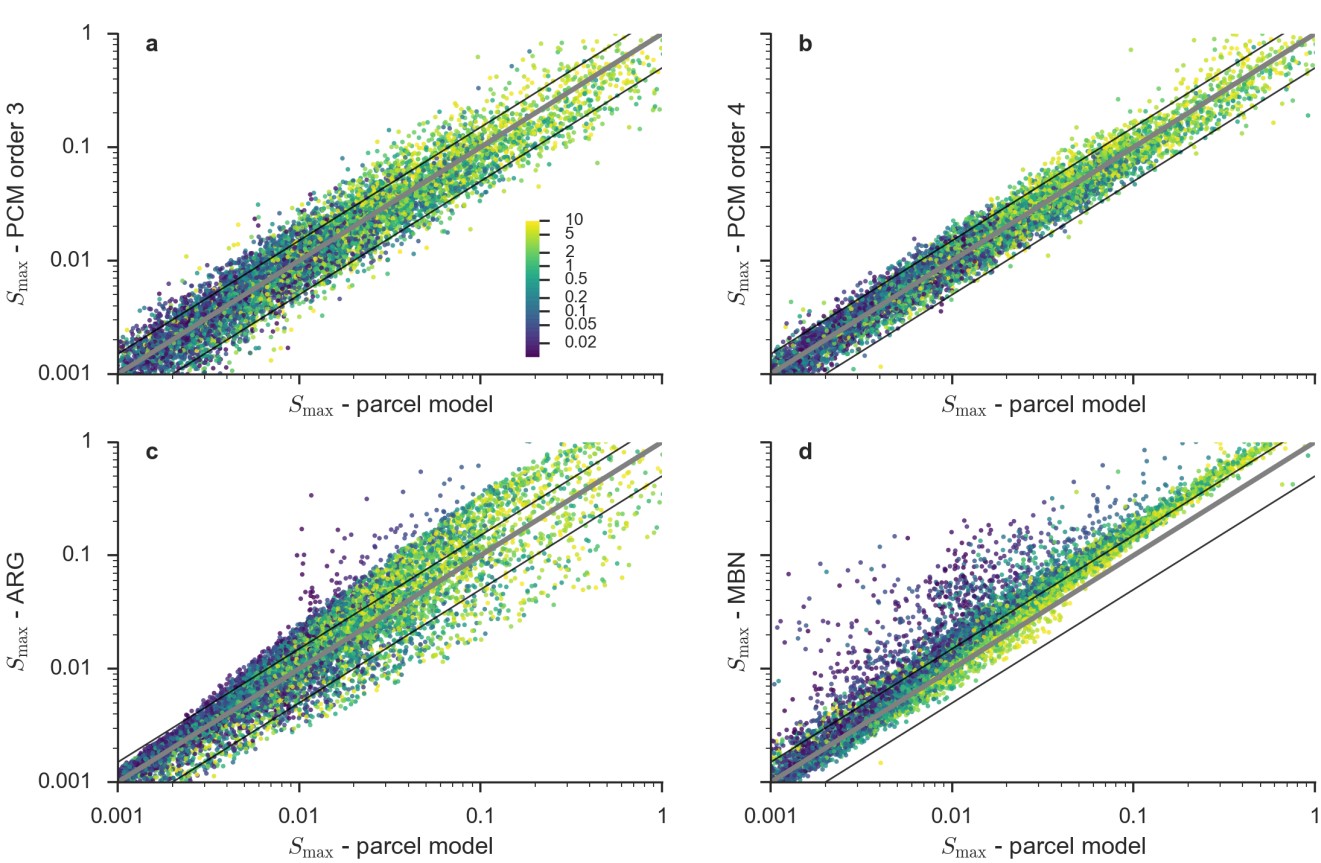

**Figure 7.** Like Figure 5, but for the "gCCN" parameter set



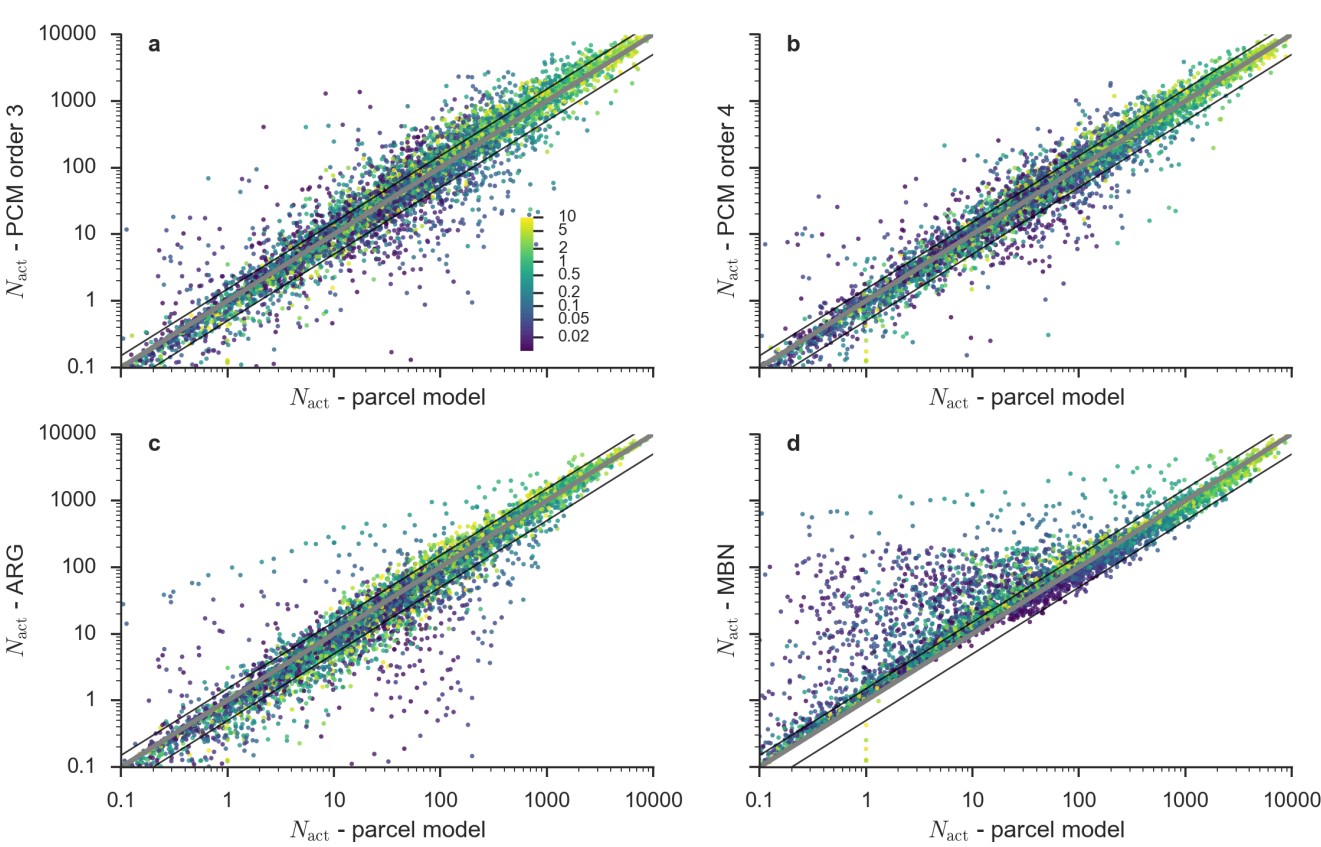

**Figure 8.** Like Figure 6, but for the "gCCN" parameter set



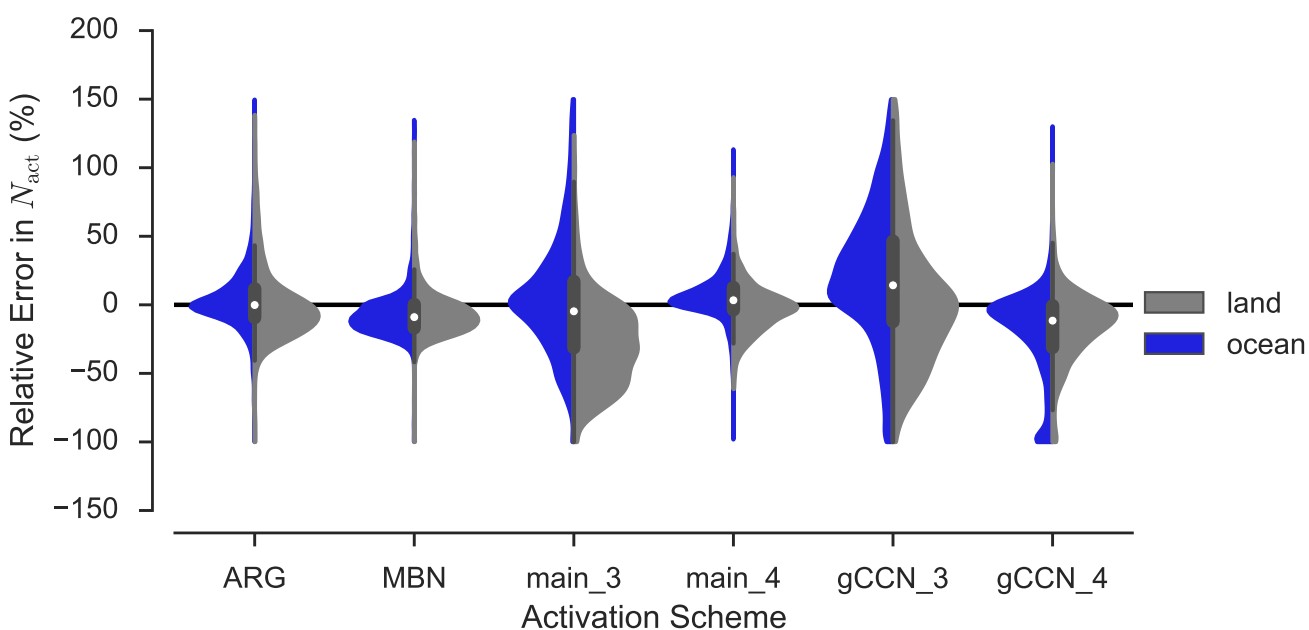

**Figure 9.** Distributions of relative error in scheme prediction of $N_{\text{act}}$ versus detailed parcel, evaluated using samples taken from instantaneous MARC aerosol size distribution and meteorology and colored by geographical regime. The long tail of each distribution is clipped at the extrema for each scheme. The box plot in the center of each distribution shows the median and inter-quartile range of the total distribution of both land and ocean samples for each scheme.







**Figure 10.** Mean relative error in scheme prediction of $N_{\mathrm{act}}$ versus detailed parcel model plotted against location on globe where those samples originated. At each grid location, all samples across timesteps and vertical levels (below 700 mb) are averaged together to compute the mean.



| Aerosol Mode | Geometric Mean Particle Diameter (µm) | Geometric Std Deviation ($\sigma_g$) | Density (g cm$^{-3}$) | Hygroscopicity ($\kappa$) |
|---|---|---|---|---|
| NUC | 0 to 0.005 84 | 1.59 | 1.8 | 0.507 |
| AIT | 0.005 84 to 0.031 | 1.59 | 1.8 | 0.507 |
| ACC | >0.031 | 1.59 | 1.8 | 0.507 |
| OC | - | 2.0 | 2.0 | $10^{-10}$ |
| MOS | - | 2.0 | † | † |
| BC | - | 2.0 | 2.0 | $10^{-10}$ |
| MBS | - | 2.0 | 2.0 | 0.507 |
| DST01 | 0.16 | 1.4 | - | 0.14 |
| DST02 | 0.406 | 1.4 | - | 0.14 |
| DST03 | 0.867 | 1.4 | - | 0.14 |
| DST04 | 1.656 | 1.4 | - | 0.14 |
| SSLT01 | 0.5 | 1.59 | - | 1.16 |
| SSLT02 | 2.0 | 1.37 | - | 1.16 |
| SSLT03 | 5.0 | 1.41 | - | 1.16 |
| SSLT04 | 15.0 | 1.22 | - | 1.16 |

**Table 1.** MARC aerosol mode size distribution and composition parameters. The MOS mode (†) has a composition-dependent density and hygroscopicity which is computed using the internal mixing state of organic carbon and sulfate present at a given grid-cell and timestep.



| Symbol | Description | Lower Bound | Upper Bound |
|---|---|---|---|
| logN_ACC | Log of accumulation mode sulfate number concentration ($cm^{-3}$) | -3 (1.2) | 4 (100) |
| logN_MOS | Log of mixed sulfate-organic carbon number concentration ($cm^{-3}$) | -5 (1.5) | 4 (99.9) |
| logN_MBS | Log of mixed sulfate-black carbon number concentration ($cm^{-3}$) | -5 (1.6) | 4 (99.8) |
| logN_DST01* | Log of 0.16 micron dust particle number concentration ($cm^{-3}$) | -5 (18.2) | 2 (99.8) |
| logN_DST02* | Log of 0.4 micron dust particle number concentration ($cm^{-3}$) | -5 (38.9) | 1 (99.9) |
| logN_SSLT01* | Log of 0.5 micron sea salt particle number concentration ($cm^{-3}$) | -5 (3.6) | 1 (100) |
| logmu_ACC | Geometric mean size of accumulation mode (micron) | -3 (0.1) | 0 (98.9) |
| logmu_MOS | Geometric mean size of mixed sulfate-organic carbon mode (micron) | -3 (0.06) | -1 (98.3) |
| logmu_MBS | Geometric mean size of mixed sulfate-black carbon mode (micron) | -3 (0.1) | -1 (98.5) |
| kappa_MOS | Hygroscopicity of mixed sulfate-organic carbon mode | 0.1 | 0.6 |
| log_V | Log of updraft velocity (m/s) | -2 | 1 |
| T | Temperature (K) | 240 | 310 |
| P | Pressure (Pa) | 50000 | 105000 |

**Table 2.** Input parameter space and bounds on associated uniform probability density functions used to derive polynomial chaos expansions for MARC activation. For the lower and upper bounds on the aerosol size distribution parameters, the parenthetical values denote the percentile of the distribution for that parameter at which the bound occurs. All terms are present for the main expansion; terms affixed with an ($^{*}$) are added for the gCCN expansion.

| | | $\log_{10} S_{\max}$ | | | | $N_{\mathrm{act}}$ | | | |
|---|---|---|---|---|---|---|---|---|---|
| exp | scheme | MAE | MRE | NRMSE | $r^2$ | MAE | MRE | NRMSE | $r^2$ |
| main | ARG | 0.18 | -3.26 | 0.10 | 0.94 | 40.14 | 25.39 | 0.15 | 0.98 |
| | MBN | 0.20 | -11.79 | 0.18 | 0.81 | 59.05 | 44.95 | 0.30 | 0.90 |
| | PCM Order 3 | 0.16 | 0.59 | 0.09 | 0.95 | 72.54 | 9.20 | 0.31 | 0.90 |
| | PCM Order 4 | 0.10 | -0.60 | 0.06 | 0.98 | 45.47 | 8.89 | 0.19 | 0.96 |
| gCCN | ARG | 0.17 | 8.54 | 0.09 | 0.93 | 37.41 | -3.92 | 0.15 | 0.98 |
| | MBN | 0.20 | -9.58 | 0.17 | 0.78 | 56.03 | 33.30 | 0.31 | 0.89 |
| | PCM Order 3 | 0.16 | 0.59 | 0.08 | 0.95 | 81.19 | 15.14 | 0.34 | 0.87 |
| | PCM Order 4 | 0.10 | 0.36 | 0.06 | 0.98 | 50.99 | 6.90 | 0.23 | 0.94 |

**Table 3.** Summary statistics for error in supersaturation maxima and droplet number nucleated predicted by emulators and activation parameterization relative to corresponding simulations with a detailed parcel model. From left-to-right, each column represents the coefficient of determination ($r^2$), mean absolute error (MAE), mean relative error (MRE), and the normalized root-mean-square error (NRMSE)

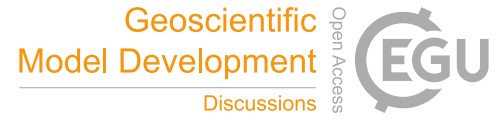

| scheme | $\log_{10} S_{\max}$ | | | | $N_{\mathrm{act}}$ | | | |
|---|---|---|---|---|---|---|---|---|
| | MAE | MRE | NRMSE | $r^2$ | MAE | MRE | NRMSE | $r^2$ |
| ARG | 0.05 | -0.16 | 0.03 | 0.92 | 25.5 | 2.87 | 0.16 | 0.94 |
| MBN | 0.06 | 0.05 | 0.05 | 0.71 | 26.7 | -6.68 | 0.19 | 0.93 |
| main Order 3 | 0.12 | 0.42 | 0.08 | 0.33 | 64.7 | -1.81 | 0.44 | 0.59 |
| main Order 4 | 0.04 | -0.31 | 0.02 | 0.96 | 24 | 4.59 | 0.19 | 0.93 |
| gCCN Order 3 | 0.14 | -1.84 | 0.09 | 0.18 | 75.6 | 20.9 | 0.47 | 0.52 |
| gCCN Order 4 | 0.12 | 4.61 | 0.10 | -0.19 | 44.3 | -19.9 | 0.33 | 0.76 |

**Table 4.** Same as Table 3, but for the sampling study using MARC aerosol and meteorology parameter sets.