# Peer review of "An aerosol activation metamodel of v1.2.0 of the pyrcel cloud parcel model: Development and offline assessment for use in an aerosol-climate model"

_Geoscientific Model Development, 2016_

## Referee Comment (RC1) · Anonymous Referee #1 · 29 Nov 2016

This research is a follow-up from an earlier work (Rothenberg and Wand, JAS, 73, 2016) in which an emulator of an aerosol activation parcel model is built. The emulator is built by utilizing a polynomial chaos expansion. In the current work, the authors evaluate the emulator of the aerosol activation process by using instantaneous aerosol and meteorological fields from the MARC model to drive the emulator. The evaluation is performed by comparing the emulator predictions (Maximum Supersaturation - Smax, and Number of activated aerosol particles - Nact) against the detailed parcel model simulations, as well as against commonly used activation parameterizations.

[Figure]

The work shows that when the space of input parameters is sampled in the full range of potential combinations, which are also used to train the emulators, the performance of the emulators seems to be better than those of the activation parameterizations. However, when realistic input is used, the performance of the emulators is deteriorated and the activation parameterizations perform better, with lower standard deviation of the errors.

General Comments:

The manuscript presented here presents an interesting and somehow novel approach at generating emulators of aerosol activation schemes. The authors claim that the approach could be implemented in aerosol cloud interaction studies with climate models, and that it could be a good way of dealing with the increasing complexity of the aerosol schemes used in state-of-the-art global climate models. However, some serious conceptual problems are observed in several points during the manuscript, all of which need to be revised before the manuscript is suitable to be published. Main problems with the document are:

- The document has a lengthy historical introduction to the study of aerosol activation process. However, there is abundant literature on this topic, including review papers. This paper is basically a methods paper, in which a different approach is used to built and evaluate an emulator. As such, that should be the focus of the paper and should be written accordingly.

- Despite being focused on the description of the emulator development and evaluation, the document fails to discuss other, similar studies, based on the concept of a statistical emulator, eg., Patridge et al., Atmos. Chem. Phys., 11, 7269–7287, 2011, Carslaw et al., Nature 2013. . . etc.). - The paper is unnecessarily long, especially given the fact that it is a follow up paper and most of the details of the emulator development are described in an earlier work. It should be much more concise and focused on the evaluation procedure.

- A reference is made in the abstract to GCCN, but no explanation for this is given. Is the focus of the study the analysis of GCCN impact on accuracy of the prediction?

- It is unclear what is the potential use the emulator presented here could have. The manuscript deals with offline evaluation of the emulator, but no plan or path forward for its online implementation and evaluation is mentioned nor discussed. The authors do not discuss the computational cost associated with the emulator compared to in-use parameterizations.

- Contrary to what the authors claim in the paper, activation schemes have been tested using complex aerosol distributions from state-of-the-art GCMs and chemical transport models. The activation parameterizations are not only tested with idealized aerosol size distributions.

Specific Comments

- The introduction should be made a lot more concise. The authors include here a very long historical review of activation parameterizations, but it could and should be done in a much more concise way. L33-page 3. "many evaluations of activation scheme performance have focused on the same set of relatively simple aerosol particle size distributions". This sentence is inaccurate. Many evaluations of activation parameterizations have been performed using aerosol fields produced from global aerosol models (such as MAM-3 from Liu et al., 2012), not only with idealized aerosol fields.

- I strongly encourage the authors to re-write some portions of the paper. Section 2, Activation Parameter Space, should be shorter. Use of a table (such as table 1) and a short explanation in the body of the manuscript should be enough to describe the aerosol populations produced by MARC.

- Line 19-page 6. "The different particle size distributions in MARC are influenced by both different emissions sources and acted upon by different physical processes. This leads to a great deal of spatial heterogeneity in the size distribution parameters.

[Figure]

One aspect of this heterogeneity is depicted in Fig. 2,….." — This is a characteristic common to almost all current aerosol models used in general circulation models. Such general sentences should be removed.

- The usefulness of Figure 2 is not clear. I suggest removing it or using a more effective visualization of what is being said. For instance, a few global maps showing regional patterns or zonal averages could be utilized here instead.

- Section 2.2. Could be merged with section 2.3. This initial "reduction of parameter space" could be explained in just a few lines. It some modes have no hygroscopicity whatsoever, then, they cannot affect supersaturation balance, and will not be activated.

- Section 2.3, should be re-written. Although the authors presented a lengthy introduction to the aerosol activation problem, this section largely ignores what has already been learned in the topic for the last few decades. It is well known that nucleation mode particles are not important for activation (too small, too high of a critical supersaturation), coarse particles are important in suppressing supersaturation development (large, few, but kinetically limited) but not great contributors to the number concentration. Authors should refer to these facts, but there is no need to use 2 pages to explain the iterative procedure.

L31-p7 – "This strategy effectively employs a "greedy" algorithm to sort the set of available aerosol modes, ranking their influence on activation by their cumulative depression on the supersaturation maximum achieved for a given parcel ascent." – Consider removing this sentence since its redundant with the next paragraph.

L13-p8 "Thus, it would be reasonable to assume that the nucleation mode particles would "dominate" activation in this case - or that small changes in the burden of these particles could have large consequences on how many cloud droplets will form. . . . However, that doesn't happen." I would suggest to the authors removing this kind of sentences from the manuscript. In the lengthy introduction to the paper, the authors already explained some of the basics of the activation process. Only if the process

had never been studied would it be "reasonable" to assume nucleation mode particles would play an important role in activation. It is well known this is not the case.

- The need for the paragraphs from line 10 to line 30 in page 8 is unclear. I would suggest removing or rewriting all the sentences from "The vast majority of particles exist..." to "....which leads to a more rapid balancing between the parcel's adiabatic cooling and warming due to latent heat release". The latter sentence is completely inaccurate and reflects an incomplete understanding of the activation process. During the development of a maximum supersaturation adiabatic cooling is never balanced by the latent heat release. The parcel keeps on cooling as long as there is an updraft forcing it upwards. The latent heat release slows down the cooling rate, but what really controls the maximum supersaturation is the mass transfer of water from the vapor to the liquid.

L10-p9 "This is consistent with the physics of the activation problem; the presence of more aerosol surface area on which condensation can occur tends to produce a greater source of latent heat release to counter-balance adiabatic cooling in the ascending parcel, suppressing the development of a higher Smax" —- Same conceptual mistake. The phrase is incorrect or at least inaccurate. In aerosol activation, the reduction of water vapor is by far more important in reducing the rate of increase of supersaturation, than the release of latent heat.

- The same conceptual mistake is made again in lines 7 and 15 in page 10.

- Line 15 – Page 12. Authors should discuss in much more detail the implementation limitation of their proposed method. " The trade-off here potentially lies in emulator performance, because the PCM will attempt to train the emulator to perform well for input parameters that we've assigned equal likelihood to, but may actually be far less likely to occur." This kind of drawback seems to be shared between all the statistical models used, which are also discussed by the authors in the introduction.

- The authors devote too much time to the discussion of the emulation in the entire

range of mathematically (but highly nonphysical) combination of parameter values. Figures 5, 6, 7, and 8 are used to illustrate the results for the entire range of potential set of input parameters. Most of the discussion of the discrepancies between parcel model simulations and activation parameterization is focused on regimes unlikely to be found (e.g., extremely clean environments with less than 1-10 #/cm3). Some other emulator studies have used expert elicitation to constrain the ranges of some parameters, so they can sample physically meaningful regions of the parameter space.

---

## Referee Comment (RC2) · Anonymous Referee #2 · 20 Dec 2016

This study describes the development and application of an emulator for aerosol activation in an aerosol-climate model. Given that aerosol/cloud interactions are among the largest uncertainties in current estimates of radiative forcing, any progress in their accurate representations in global models is welcome. The current study certainly contributes to this; however, several changes are needed in order to streamline and clarify the manuscript.

Major comments

1) Structure/Length of the manuscript

[Figure]

The manuscript is rather lengthy and reads in several section like a thesis, including a lot of 'common knowledge' material and information that can be found in standard textbooks. Many sections can be shortened so that the main focus of the study is more pronounced and trivial information is removed.

a) For example, in the introduction many activation studies are described that consider effects such as surface-tension reduction or kinetic growth that are irrelevant to the current study.

b) Another example is the lengthy description of the iteration finding the dominant modes (Section 2.3). There are many sensitivity studies that have shown that size is the most important parameter influencing CCN (and therefore droplet) number. Therefore, the discussion of the small role of the nucleation mode is redundant.

c) In general, several parts are repetitive and can be merged or omitted as they are trivial and not new. One example is the beginning of Section 3.3 (p. 11, l. 23-30) that can be summarized in one sentence. Another one is the description of the sensitivity test at the end of Section 2.3 which are a direct result of Koehler theory.

2) Comparison to other activation schemes

The comparison of the emulator to the other aerosol activation schemes seems a bit 'unfair', or at least not clear. The activation schemes by Abdul-Razzak/Ghan (ARG) and Morales Betancourt/Nenes (MBN) were developed based on fits to parcel model simulations. Only if all the initial parcel models use the same input parameters, e.g. giant CCN (gCCN), the fits will be appropriate in matching such extreme situations. The text on p. 18, l. 14 suggests that this was not the case but this should be brought up earlier.

3) Features of the aerosol size distribution in MARC

It seems that the MARC model includes many features that are not explicitly described here but might have an impact on the aerosol activation. For example, the variable hygroscopicity of the MOS mode is mentioned in Section 2 (p. 5, l. 16) but the processes that lead to it are not further described. What is the extent to which hygroscopicity changes? How does it compare to treatment in other models?

4) Previous work by the same authors

It is clear that the current study is a follow-up study of the previous study by the same authors (Rothenberg and Wang, 2016). However, without reading this previous paper, it is not evident which parts are actually new here and which is merely a repetition of the previous work. Clarifying this would also likely help to shorten and focus this manuscript.

5) Role of gCCNs

There are several prior studies that explored the role of gCCNs for cloud properties, and also highlighting that they have largest effects in clean environments, e.g. Feingold et al., J. Atmos. Sci., 1999; Yin et al., Atmos. Environm., 2000). Therefore, the results in Fig 10 are not that surprising (p. 17, l. 7) and should be accordingly put into context of earlier studies.

6) Applicability of the emulator

a) It is shown that the emulator performs worst for extreme cases, e.g. for very clean conditions. The reason for this is ascribed to the fact that it was not trained for these conditions. However, it seems trivial to extend the parameter range of the parcel model to cover such conditions. On the other hand, some more discussion should be given for the reasoning of less extreme conditions and therefore the applicability of the emulator for most conditions globally.

b) The current study represents the next step of the previous study, namely the implementation of the emulator in an aerosol climate model. However, the computational benefits should be discussed more in comparison to previous activation schemes.

Minor comments

p. 1, l. 3-4: Many recent model studies have aimed at reducing this 'ever-increasing complexity' by reducing the number of bins and/or making simplifying assumptions on aerosol composition (e.g., kappa Koehler theory, omission of surface tension reduction as it has been shown to be negligible etc).

p. 2, l. 17; and p. 18, l. 35: 'processes' does not seem the right word here since e.g. surface tension reduction is not a process but rather an effect.

p. 6, l. 18: What is the value of the artificial cap?

p. 8, l. 30, and several other places: The condensation of water vapor onto particles/droplets is the main factor that occurs during the adiabatic ascent of an air parcel. The latent heat release is only a minor driving force of the condensation growth.

p. 10, l. 18: What are 'proto-cloud droplets'?

p. 10, l. 19, and l. 27: 'hydrated' usually refers to a chemical reaction where water is a reactant and its reaction results in a different product (e.g. hydrated aldehydes). Better here, use 'interstitial haze particles which have taken up some water according to their hygroscopic mass' (or similar).

p. 12, l. 25/6: This statement requires an explanation. Does it imply that kinetic limitations on growth bias the 'true results' and therefore should not be included?

p. 15, l. 29: How can a model underpredict by more than 100%? Wouldn't that result in a negative concentration? Clarify.

p. 17, l. 29: 'a chain of physics which ultimately leads to the aerosol indirect effect on climate' is very colloquial and vague. Rephrase.

p. 18, l. 27: It is not clear here what Gantt et al compared causing a difference of -0.9 W m-2.

p. 19, l. 1: 'myriad facts' is very vague and should be specified.

p. 19, l. 5: This is the first time in the manuscript that it is mentioned that there is a discrepancy between satellite-derived and model-predicted cloud droplet number. In order to pose the problem, this should be mentioned in the introduction, together with appropriate references.

Figure 1: In order to show the scale more clearly, you might consider a gap on the y-axis (e.g. from 2-6)

Figure 3: The figure seems very trivial and repeats text book knowledge. In the process of shortening the manuscript, it can be omitted and just described by a sentence.

Figure 4: Define DST01, MBS, MOS and ACC in the caption.

Figure 5: 'One-one plot comparing' can be omitted.

Technical comments

p. 2, l. 2 and at several other places in the manuscript: 'Aerosol' is singular and should be followed by a verb in singular form. Alternatively, use 'aerosol particles' which is more accurate anyway.

p. 2, l. 18: replace 'change sin' by 'changes in'

p. 2, l. 19: Not clear what 'their' refers to – to 'parcel model calculations' or 'global models'

p. 3, l. 10: Remove 'J.-M.'

p. 5, l. 6: Aitken

p. 5, l. 17: replace 'forms' by 'form'

p. 5, l. 25: 'from aerosol' is redundant

p. 6, l. 33: define 'u(g)' here

p. 10, l. 10: If you decide to include these equations (which is not necessarily needed;

a simple reference to a standard text book such as Pruppacher and Klett would be sufficient), write them as equations in separate lines.

p. 12, l. 21/22: The last part of this sentence seems incomplete; 'activate' might be missing at the end.

p. 15, l. 34: Period missing after 'range'.

p. 17, l. 23: a mean relative error

Reference list: I believe that GMD does not require the full URL for papers when the DOI is listed.

Morales Betancourt and Nenes, 2014b should be cited as their GMD (not GMDDiscussion) paper.

---

## Referee Comment (RC3) · Anonymous Referee #3 · 21 Dec 2016

The authors show how their previously published methods for developing an emulator that approximates 1D parcel model output can be extended to include the parameter set used by a full aerosol model. Results of the derived emulator are compared against parcel model output for a large range in possible parameter values and for more realistic parameter value sets from model data. The emulator performance compares well to existing parameterizations with smaller biases in maximum supersaturation and activated particle number for the full range in parameter values, but compares less well when the parameter sets are taken from model data.

[Figure]

My overall impression is that this new approach to parameterizing droplet activation shows promise for global modeling and could be of interest to those in the GCM cloud microphysics community. If indeed this is directed toward these interests (as suggested by the title/abstract) the manuscript could use more information on aspects of the scheme that are relevant for global modeling, notably computational expense, and also more of a focus on conditions and aerosol populations characteristics that are relevant to global modeling (i.e. CDNC > 10cm-3). I also have some questions about some of the choices made regarding initial parameter values and some of the simplifications made. My recommendation is for major revisions, including potentially to some aspects of the experiments and tests, in accordance with my comments and suggestions given below.

General comments:

1. One aspect of the emulators that is not mentioned in the text is the computational expense of this scheme relative to widely used parameterizations, such as the two used for comparison here, or the lookup table method. For use in a global model, the cost of computation would be considered along with the accuracy of the scheme. This is written about in Rothenberg and Wang (2016) and should be included here too, either by doing a quick assessment for this study, or by citing the previous work.

2. Since this is an extension of a previous paper but not a case where the code developed previously has been implemented in a global model, it is important to be clear and up front about what is new in the current study. On Pg 4 Lines 18-20 it is stated that the current study makes the Rothenberg and Wang (2016) emulator suitable for a global model, but I think it is not clear what that means at this point in the study. A bit of clarification would be helpful.

3. In general the paper seems unnecessarily long, especially given that it is an extension of previous work. I would recommend as a strategy for being more concise to find parts of the text that are common to this paper and Rothenberg and Wang (2016) and

then just refer the reader to the latter paper for details. So this would include, for example, the description of the parcel model and the polynomial chaos expansion. Another area that could be cut down is the introduction and the descriptions of all the different parameterization methods. Since the goal of this paper is to prepare the emulator for a global model, it should suffice to list which GCMs use which activation schemes and the advantages/disadvantages of each.

4. I am somewhat uncomfortable with the exclusion of several of the aerosol modes and other parameters from the development of the emulator (Section 2.2) without having tested the sensitivity of the activation to these parameters, as is done for the remaining parameters in the following section anyhow. I would also note that no papers are cited to justify excluding these parameters. My recommendation is to either test all the parameters in the iterative calculations, or provide justification for excluding parameters from the literature. For example, it is reasonable to assume that nucleation mode particles are too small to contribute to CCN numbers, but it is not as obvious to me that a large population of nucleation mode particles could not have an impact on Smax by depleting available water vapor. In fact, there seems to be some evidence for this in the example given in Sect. 2.3? Maybe I am mis-reading this, but in any case please include a citation(s) to justify this exclusion or run the tests with this mode included. As a side note, it can be confusing that the three modes tested in the iterative calculations are "nucleation, accumulation and coarse" when the MARC modes are named "nucleation, Aitken and accumulation" and the nucleation mode has been excluded from the analysis. I think this requires some additional explanation.

5. Also with regard to the exclusion of parameters from the emulator, the use of a value of hygroscopicity equal to that of pure sulfate for the sulfate/BC mixture does not match up with my understanding of how the kappa parameter is usually applied. My understanding is that kappa is computed as a volume-weighted average of the component species regardless of the arrangement of the species within the particle. One could conceive of a situation in which this would not work, namely hygroscopic

material within a surrounding shell of insoluble material, but I think this is considered rare. For the opposite, more common arrangement (hygroscopic with insoluble core) the volume-weighted kappa is appropriate, especially in this case since sulfate is highly soluble. So the water activity of a half:half BC:sulfate particle would be much less than the pure sulfate as is assumed here and I suspect the activation within the parcel model/emulator is sensitive to this for certain conditions. Please revisit this and see if you agree.

6. With regard to the choice of parameter space to cover - I understand the idea to cover a large range of parameter values for the sake of completeness. However, I find it curious that the lower bound for the geometric mean sizes is 1nm for all modes after using the small size of the nucleation mode particles to justify excluding them altogether from the analysis. Perhaps since the values are sampled from a uniform distribution these small values do not come up very often and so the lower bound does not matter very much? And in this case does the sampling overemphasize higher values of, for example, ACC sulfate number concentration which ranges from roughly 0 to 10,000 /cm3 and therefore could be expected to be > 5000/cm3 about half the time? I think my major question here is: do you have a sense for how sensitive the emulator is to the parameter set used to train it? Could making reasonable changes to the range or sampling of parameters make a big difference in emulator performance? If so, it might be worth noting that many GCMs set a minimum CDNC, often 10/cm3 or greater, and this might direct future emulator training efforts.

Minor comments:

Pg 2, Lines 14-18: This paragraph begins by describing ESMs and moves on to describing parcel models without a clear transition. I suggest starting a new paragraph beginning with Line 14.

Pg 2, Line 19: My understanding is that the barrier to including a parcel model within a GCM is solely related to computational expense. Otherwise there would be no need to

parameterize or emulate the parcel model.

Pg. 4, Line 33: I suggest noting, as you do, that MARC can be used instead of MAM3 or MAM4 in CAM5 (I am not sure it replaced MAM3?), but remove the wording that MARC "extends" CESM. Is there a reference for the implementation of MARC in CAM5?

Pg 5, Lines 23-25, 28-30: I may be just confused about what MARC is – does it also include cloud microphysics?

Pg 5, Lines 32-33: I wonder what impact this assumption has on the results especially since V is lower than the set limit 50% of the time. Is there a reference that could be used to justify making this assumption or maybe some tests of your own?

Pg 6, Line 11: Please note and cite the emissions datasets that are used, particularly for dust and sea-salt.

Pg 6, Line 33: I think mu-sub-g needs to be defined.

Pg 7, Lines 1-7: This paragraph could probably be deleted in the interest of space.

Pg 7, Lines 18-19: For practical purposes this is roughly true, but in theory organic carbon aerosol are not all insoluble or non-hygroscopic and even insoluble particles can activate droplets at high enough SS if they are large.

Pg 7, Lines 21-22: Is this assuming a fixed size range for the nucleation mode? Later on it is shown that the accumulation mode mean size covers a huge range and I wonder if it is conceivable that a variable nucleation mode mean size could vary up into sizes that are more relevant for CCN?

Pg 8, Lines 26-30: I have difficulty interpreting the reductions in Smax by including the nucleation or coarse modes as small or large effects. Is it possible to report the changes in activated droplet number instead? This might be easier to understand for other readers as well. I would add that the reductions in Smax are also due to the decrease in available water vapor in the case with more condensation.

Pg 8, Lines 31-32: Could you provide more details about these 50,000 parameter sets? Were they sampled randomly for a range in altitudes?

Pg 9, Lines 11-13: I have to admit that I do not know the relative impacts of the reduction in parcel buoyancy vs. the reduction in water vapor for suppressing the Smax. Is this something you have looked into? Might be worth mentioning both mechanisms here in any case.

Pg 12, Lines 13-15: Is this speculation or is this something that was observed from the MARC output? Several major biomass burning areas (large sources of BC) are coastal or near-coastal such as in Australia and Indonesia and large indirect effects from biomass burning aerosols have been deduced from modeling off the west coast of Africa and South America. I would expect a mix of BC, sulfate and sea-salt in these places.

Pg. 12, Line 25: I would define "parcel model behavior" or use a more specific phrase such as "parcel model Smax"

Pg 13, Line 19: This sentence was difficult to understand, perhaps just changing the word choice for "case" would help clarify.

Pg 14, Lines 8-13: I would argue that the bias for simulations with CDNC > 10/cm3 are of more interest to the global modeling community – maybe this discussion could be removed from the text.

Pg 15, Lines 32-33: I was surprised to read that the aerosol number concentrations ever went below the thresholds used to train the emulators which were 1e-5 /cm3 for all except ACC mode sulfate (1e-3 /cm3). Is the model often predicting values this small? It is possible that I misread this.

Figure 5-8: Please include units for figures 5-8 (% and /cm3)

Table 2: Are the hygroscopicity range values rounded off? Kappa should not be above the value for sulfate and could drop well below 0.1, although maybe sulfate usually

[Figure]

dominates the internal mixture volume.

---

## Author Comment (AC1) · 3 Feb 2017

Response to reviews and comments on "*An aerosol activation metamodel of v1.2.0 of the pyrcel cloud parcel model: Development and offline assessment for use in an aerosol-climate model*"

We would like to think the reviewers for their thoughtful, thorough, and helpful suggestions and feedback on our manuscript. We appreciate the reviewers' recognition of the potential for emulator approaches, such as the one we discuss in our manuscript, to improve parameterizations of aerosol-cloud interactions in global models, and have made efforts (as documented below) to address the additional questions and issues they identified pertaining to the manuscript.

Below, we respond to each of the "General" and "Specific" comments from the three reviewers, and recorded the actions taken in response to the "Minor" comments provided. We first respond generally to three additional, common "General" comments raised by all three reviewers here:

1. All three reviewers requested additional details on the computational expense of the metamodels, and more information on how they could be used in global models. Up to and during the review of this manuscript, we completed the implementation of a set of our activation metamodels into a global climate mode. Although we defer the discussion of the full impacts of these schemes on the simulated climate to a future manuscript currently *in preparation*, we are able to include a discussion of the impact the emulators have on the computational expense of the global model, compared with the other activation schemes we consider in this manuscript. This discussion has been added to the concluding section of the manuscript. In short, 3 of the 4 activation models yield negligible impact on the performance of MARC versus the reference ARG implementation. The MBN scheme increases the computational expense by about 7%; the 4th-order gCCN scheme incurs an increase of just 3%.

2. All three reviewers expressed concern that the manuscript was much too long, did not explicitly clarify its differences with Rothenberg and Wang (2016), and did not focus narrowly enough on the develop and evaluation of the metamodels. To address these points, we have re-written the Introduction nearly in its entirety to exclude historical details of activation parameterization development better recounted elsewhere (e.g. in Ghan et al. (2011)), and consolidated the discussion in Sections 2 and 3 into a single section entitled "Emulation of aerosol activation". Furthermore, we have worked to eliminate the reproduction of material available in Rothenberg and Wang (2016), opting to refer the reader there for details on the parcel model used here and on the polynomial chaos expansion technique.

3. All three reviewers included comments on redundancies in Section 2 and the iterative activation calculations. As mentioned previously, we have consolidated this Sections 2 and 3 and streamlined the discussion in both. Where possible, we have tried to include additional references to justify our decisions on how to exclude particular aerosol modes. Overall, we still feel it is important to include this discussion in the manuscript; a key difficulty in employing emulators and other statistical approaches to study physical processes and to develop parameterizations is the "curse of dimensionality," where increasing the number of important parameters tends to greatly increase the expense of building and evaluating a given emulator or statistical model. A critical component of such an effort, then, is rigorously searching for reduced sets of parameters to use in building the statistical models, often referred to as "feature extraction" in the machine learning community or otherwise "dimensionality reduction". Although there is a large literature on algorithms and automatic ways to accomplish this, these techniques often use a brute-force or exhaustive search approach. Our iterative calculations are the result of attempting to build a physically-informed algorithm for approaching feature extraction, and we hope that such an approach will be informative for researchers who pursue similar lines of study in the future. Therefore, we have tried to retain at least a limited discussion of the iterative calculations and results in the revised manuscript.

The remaining reviewers' comments are addressed below. We have made our best effort to address each individual comment in the sections labeled by the reviewers (General/Specific/Major/Minor/Technical comments, where denoted), and in the same order/numbering as they were presented. For clarity, we have included a summary of the reviewer's comment **in bold** before each response. Although this resulting document is somewhat long, we hope it fully documents our attempts to incorporate all of the thoughtful critiques provided by the reviewers and outlines our planned revisions to the manuscript.

**Reviewer #1**

General Comments

- **Lengthy historical introduction**

  We have opted to extensively re-write the introduction to streamline the introduction, referring the reader to other literature for more information. The new introduction is focused on the critical literature most pertinent to the development of our new method and parameterization.

- **Failure to focus on description of emulator development and related work**

  We have included discussion of the noted literature (Partridge et al. (2011) and Carslaw et al. (2013)) and other work on emulation as applied to atmospheric and climate processes, and made an effort to place this new work in that appropriate context. We have also worked to reduce the duplication of details already available in Rothenberg and Wang (2016).

- **GCCN**

  Although the focus of this manuscript is not on analyzing the impacts of GCCN on the accuracy of activation calculations in general, we did wish to understand how their inclusion/exclusion could influence the relative performance of our metamodels. We agree that this was not sufficiently introduced in the manuscript, and have added a brief discussion on how the presence of GCCN impacts aerosol activation in the introduction. Additionally, we added some clarifying remarks where the separate "main" and "GCCN" schemes are introduced in the manuscript to better highlight our motivation for training two separate emulators.

- **Path forward/computational cost**

  We defer the reviewer to our comments at the beginning of this document. We also include brief remarks in the conclusion of the manuscript discussing our implementation in the global model and its implications for future work.

- **Testing of activation schemes**

  We agree with the reviewer that, in fact, activation schemes have been tested against complex aerosol distributions from a variety of tools, including GCMs and CTMs. However, we stress that our remarks about testing the schemes refers more-so to inter-comparisons of different

activation parameterizations. Since the comprehensive review of Ghan et al. (2011), few works have looked at how different activation schemes perform when coupled to the same host model, or using the same set of complex aerosol size distributions. In order to more accurately represent the historical work on this issue, we've amended some of our comments on this topic in the manuscript, especially in the Introduction (e.g. the first *Specific Comment* below).

Specific Comments

The following responses match exactly to each of the comments provided by the reviewer, in the order provided. We have included reference line numbers (used in the review) to help keep track of which comment is which, where possible.

- We've rephrased **P3/L33** following the final *General Comment* above and the reviewer's comment
- Section 2 has been entirely re-written, and consolidated with Section 3.
- **P6/L19**: Removed
- **Figure 2**: We've revised our presentation of the aerosol size distribution parameters, and have replaced the original Figure 2 with a plot featuring PDFs of those parameters. This complements Table 2 and our discussion of the emulator approach later in the manuscript.
- **Section 2.2**: Replaced and consolidated with Section 3
- **Section 2.3**: Replaced and consolidated with Section 3. In the revised manuscript we have greatly shortened the discussion of the iterative procedure.
- **P7/L31**: Re-phrased and worked differently as part of the shortening mentioned previously.
- **P8/L13**: Removed, similarly to the previous comment.
- **P8/L10-30**: Removed, similarly to the previous comment. Reviewer #2 raised a similar concern about our discussion of the mechanism driving supersaturation production. We agree with both reviewers that our explanation is incomplete, and derives from a too-literal interpretation of our Equation (1). We've re-written the discussion (which occurs several times, as indicated in the next two comments) to emphasize the role of water vapor availability/reduction in controlling the maximum supersaturation.
- **P9/L10**: See previous comment
- **P10/L7,15**: See previous comment
- **P12/L15**: Re-worked the discussion on the limitations of our methodology in the consolidated Sections 2 / 3.
- **Figures 5-8**: We note that the discussion which includes Figure 5-8 still draws samples from the set of uniform distributions described in Table 2, which were constructed using the observed ranges of aerosol size distribution parameters sampled from MARC. However, the first sampling study presented in the manuscript assumes these distributions to be independent, which is where the potential for non-physical combinations arises. We've tried to re-balance the discussions of our results more towards the aerosol size distributions sampled from MARC.

Reviewer #2

Major Comments

1. **Structure and Length of the manuscript**

We have embraced all three reviewers' comments about the structure and length of manuscript and have undertaken substantial revisions and re-writing to address them. Specifically:

a) The introduction has been nearly entirely re-written. We've removed references to studies of processes irrelevant to the results we present in this manuscript.

b) We agree that the discussion of the small role of nucleation mode aerosol in activation dynamics is redundant. Section 2 has been consolidated and merged with Section 3. However, we have tried to retain some discussion and analysis of the iterative calculations, since we believe this procedure is more widely useful. Please refer to the comments at the beginning of this document for more information.

c) Similar to the previous comment, we emphasize that our attempt to consolidate Sections 2 and 3 addresses this issue pertaining to reporting results which are not new.

2. **Comparison to other activation schemes**

We agree with the reviewer that there is an inherent "unfairness" in evaluating the ARG and MBN schemes against our own parcel model and our sets of input parameters, and as the reviewer notes, we allude to this in the Conclusion. In the revised manuscript we bring up this point of discussion earlier. However, we do wish to emphasize that regardless of how a given scheme has been developed, to predict online CDNC in a global model, it must be coupled to some representation of an aerosol size distribution which may fall far outside the parcel model evaluations used to initially fit the scheme. As a result, it is critical to document the performance of these schemes in these extreme scenarios, which can and do arise in numerical models. We have attempted to clarify this discussion in the manuscript.

3. **Features of the aerosol size distribution in MARC**

Table 2 provides some perspective on the distribution of MOS hygroscopicity - in the model, it is restricted to vary between $10^{-10}$ and 0.507 (volume-weighted mixtures of OC and SO4). In practice, it takes a mean value of 0.27 $\pm$ 0.04 (1 standard deviation). We've clarified in the manuscript that the relative abundance of OC and SO4 dominates the small variation in $\kappa_{MOS}$. In MAM3, primary organic carbon is assumed to have a $\kappa$ of 0.1, which would tend to increase this average value, although we note that the presence of SOA, BC, and dust - all of which have smaller $\kappa$ than sulfate - would tend to reduce the average value away from that of pure sulfate. Additional analysis using a variance-based decomposition and the derived metamodels (following Rothenberg and Wang (2016)) suggests that the hygroscopicity of MOS plays a very small role in influencing droplet nucleation given our particular aerosol size distribution, and instead the size and abundance of sulfate is more important.

4. **Previous work (Rothenberg and Wang, 2016)**

We agree with the reviewer that deferring the reader to our previous study helps to streamline the present one, and have undertaken to do so while revising the manuscript. The new content here - which we emphasize in the revised manuscript - is the application to a more complex aerosol size distribution, and the training of a metamodel for use in predicting online cloud droplet number concentration in a global model which uses that aerosol size distribution.

5. **Role of gCCNs**

We've expanded our discussion of Figure 10 to put this result in the context of previous work on role of gCCN, especially in clean environments.

6. **Applicability of the emulator**

   We wish to emphasize that the emulators were in fact trained to be able to cover extremely clean conditions (see Table 2, which indicates the ranges of size distribution parameters included in the training). Figures 5 and 7 further illustrate that the emulators perform at least as well as the ARG and MBN schemes in extremely clean conditions, too. We have focused the discussion in the revised manuscript more towards the activation of aerosol size distributions sampled from MARC directly to clarify these points. We have also included a discussion of the computational performance of the schemes.

Minor Comments

1. **P1/L3-4**: We agree on this detail, and have added this caveat.
2. **P2/L17** and **P18/L35**: Changed "processes" to "effects"
3. **P6/L18**: The minimum is set to 0.2 m/s and the maximum to 10 m/s; we've tried to make this clearer.
4. **P8/L30**: This point was also raised by Reviewer #1. We've re-worded this discussion to de-emphasize the role of latent heat release and instead focus on water vapor reduction, which is a more important influence on supersaturation development.
5. **P10/L18**: Re-phrased as just "cloud droplets"
6. **P10/L19,27**: We've adopted the the phrasing recommended by the reviewer.
7. **P12/L25,26**: We've re-written this to express our intended point, which was that in Rothenberg and Wang (2016), predicting $S_{\max}$ and then diagnosing $N_{\text{act}}$ tended to be more accurate than predicting $N_{\text{act}}$ directly as an emulated response.
8. **P17/L29**: Rephrased with more specifics on how activation is part of a fundamental relationship between aerosol and CDNC, and how this influences the indirect effect.
9. **P18/L27**: Clarified that Gantt et al. (2014) used different activation schemes to simulate cloud radiative effect and in the global average, these effects differed by 0.9 W/m$^{-2}$ depending on the scheme used.
10. **P19/L1**: Rephrased sentence to avoid this vagueness.
11. **P19/L5**: We agree that this point should be made much earlier in the manuscript, and have included it towards the end of our revised Introduction as a motivation for further development of activation schemes.
12. **Figure 1**: Added a gap on the y-axis as recommended, and revised figure aesthetics.
13. **Figure 3**: We've adopted this recommendation and replaced the figure with a simple description in our revised Sections 2/3.
14. **Figure 4**: Labels are now defined in the caption.
15. **Figure 5**: Removed "one-one plot comparing" from caption.

Technical Comments

1. **P2/L2**: Included this grammatical change.
2. **P2/L18**: Corrected
3. **P2/L19**: Sentence does not appear in revised Introduction.
4. **P3/L10**: Fixed in BibTeX file with references.
5. **P5/L6**: Corrected
6. **P5/L17**: Corrected

7. **P6/L33**: Added definition
8. **P10/L10**: Equations are removed in revised Sections 2/3 in lieu of a reference to Rothenberg and Wang (2016) and standard texts
9. **P12/L21-22**: Corrected. "Activate" was missing
10. **P15/L34**: Corrected
11. **P17/L23**: Rephrased to avoid awkwardness
12. **References**: We use the provided BibTeX bibliography format and will consult with the editor on an appropriate course of action; Updated the Morales Betancourt and Nenes (2014) reference.

Reviewer #3

General Comments

1. **Computational Expense**

   We've included an assessment of the computational cost of running the CESM/MARC with the different activation schemes compared in this work, and our new emulators.

2. **New developments in this study**

   In line with the previous general comment, we indicate in the manuscript that the emulators derived here are implemented in the CESM/MARC to calculate online CDNC. We further clarify that the work necessary to accomplish this implementation (documented in the manuscript) is major advance beyond Rothenberg and Wang (2016), which only attempted to apply the PCM to emulate a parcel model using idealized, single-mode aerosol distributions.

3. **Length of manuscript**

   Following the comments made at the beginning of this document, we have substantially revised the manuscript by re-writing the introduction, removing material redundant to Rothenberg and Wang (2016), and consolidating Sections 2 and 3. We defer including a table of GCMs and their activation schemes, instead referring the reader to Table 3 of Ghan et al. (2011). In lieu of a pros/cons table for each scheme, we have simply truncated the discussion of each, referring the reader again to Ghan et al. (2011) and other works for more information.

4. **Exclusion of aerosol modes**

   In revising our discussion of the iterative calculations, we have removed the example figure which was the source of aerosol mode confusion here. To the broader point about justifying exclusion of some of the aerosol modes and parameters, we disagree here. The OC and BC modes are excluded by assumption (they are hydrophobic with $\kappa \approx 0$). Reviewers #1 and #2 strongly emphasize that nucleation mode aerosol do not contribute significantly to activation, which we agree with. No other modes are immediately excluded. The basis for excluding the Aitken and larger dust and sea salt modes follows the results of our iterative calculations. We've added some discussion on this point to the consolidated Sections 2 and 3, but for brief reference here: in general, the largest dust and sea salt modes exceedingly rarely have number concentrations greater than 1 per $cm^3$, which explains why they appear as a "dominant" mode in no case of our iterative calculations. The Aitken mode in MARC is generally small (see Table 1), which limits its influence on activation, although the reviewer's comments about the nucleation (or Aitken) mode particles being large enough in number concentration to

influence $S_{\max}$ by depleting available water vapor is still relevant. Still, we believe that the results of the iterative calculations themselves provide a defensible basis for neglecting the MARC Aitken mode in the ultimate emulation parameter set.

5. **MBS hygroscopicity**

In MARC, we assume MBS is constructed as a black carbon core fully covered by a sulfate shell. Therefore, the surface of such particles is assumed to be sulfate. In MARC, MBS forms from aging of external black carbon and continually grows from sulfate condensation and coagulation. Based on model results, the core-shell mass ratio is normally sufficiently small to support the assumption that the particle takes a hygroscopicity value comparable to that of sulfate.

6. **Sensitivity of emulator to training parameter set**

We emphasize that our training dataset does not randomly sample from the distributions in Table 2; the probabilistic collocation method provides a particular algorithm for choosing the sample parameter sets, which are far more concentrated in the "center" of the high-dimensional set of input parameter distributions (that is, near the mean of each individual parameter distribution). The over-sampling of training parameter sets we choose is a compromise to include more of the very-high and very-low parameter values in the training dataset.

The resulting emulator, though, should not be very sensitive to the exact choice of parameter set ranges. Following the recommendation of Rothenberg and Wang (2016), the final emulator is constructing using a least squares fit of the training parameters and the evaluated parcel model responses using a particular polynomial basis. However, the key ingredient here is the set of parcel model responses computed from the input parameter sets, which will not change unless the parameter space is dramatically altered.

We do note, as the reviewer mentions, that some GCMs include a minimum threshold CDNC value as a tuning parameter; one reason to use activation schemes is to reduce the bias towards too-low CDNC in remote maritime regions which such a threshold aims to correct.

Minor Comments

1. **P2/L14-18**: This is removed in the revised Introduction.
2. **P2/L19**: We agree in principle, but also note that the representation of sub-grid scale processes in a GCM grid box also includes a distribution of updraft speeds, and merely an average aerosol size distribution (for instance, a grid box straddling a coastal area could include a large city and open water, and have vastly different aerosol populations in either side of the grid box depending on whether or not the flow is on/offshore). Robustly assessing activation in such a diversity of cases would require something akin to the multi-scale modeling frameworks which embed a cloud-resolving model in each GCM grid box, which we briefly mention in the manuscript.
3. **P4/L33**: Unfortunately, there is not yet a reference for the implementation of MARC in CAM5; our two references refer to the CAM3.5 implementation. We have revised the phrasing "extends".
4. **P5/L23-25,28-30**: MARC is just the aerosol model. We use the default CESM cloud microphysics.

5. **P5/L32-33**: We refer the reviewer to West et al. (2014), which includes a comprehensive assessment of the role of sub-grid scale vertical velocity on activation and cloud radiative effects, and note this in the manuscript.
6. **P6/L11**: Added emissions citations
7. **P6/L33**: Added definition
8. **P7/L1-7**: Deleted as part of the consolidated Sections 2 and 3.
9. **P7/L18-19**: We agree, but emphasize that this is an assumption and limitation in MARC - that MARC does not include a detailed representation of organic aerosol.
10. **P7/L21-22**: The NUC mode does not have a fixed size, but takes the range indicated from Table 2, which is restricted to be quite small. When larger sulfate particles form, mass and number is shifted to the larger modes, so the NUC mode can't contain CCN-relevant (by size) particles.
11. **P8/L26-30**: This discussion was simplified in the revised manuscript, and these statistics aren't discussed in the same way.
12. **P8/L31-32**: Added a note that the parameter sets were sampled from instantaneous MARC output between 70S-70N and below 500mb.
13. **P9/L11-13**: This is an interesting point, but we do not consider changes in parcel buoyancy in our parcel model simulations, which assume an adiabatic ascent.
14. **P12/L13-15**: We rephrased this entire discussion in response to another reviewer's comments, and this particular point is no longer emphasized.
15. **P12/L25**: Same as previous comment.
16. **P13/L19**: Changed sentence to read, "*Both the ARG and MBN schemes include some parameters fit to parcel model simulations conceptually similar to the one emulated here.*"
17. **P14/L8-13**: We've revised the discussion to focus more on the parameter sets sampled from online MARC simulations.
18. **P15/L32-33**: We refer to Table 2, which includes (in parentheses on the righthand columns) to percentile at which the lower/upper bounds chosen to train the model occur in the distribution of those parameters as sampled from MARC. The model does predict these very small values in some cases, although they are not physically significant.
19. **Figure5-8**: Added units to figure caption
20. **Table 2**: It's correct that $\kappa_{MOS}$ can not exceed that of sulfate (0.507), but we train the model with a slightly larger range to better resolve that upper-most limit case. However, we note that the mean value of $\kappa_{MOS}$ is 0.27 with a standard deviation of 0.04, so our range captures nearly the entire variability in its value.

References

Carslaw, K. S., Lee, L. a, Reddington, C. L., Pringle, K. J., Rap, a, Forster, P. M., Mann, G. W., Spracklen, D. V., Woodhouse, M. T., Regayre, L. a and Pierce, J. R.: Large contribution of natural aerosols to uncertainty in indirect forcing., Nature, 503(7474), 67–71, doi:10.1038/nature12674, 2013.

Gantt, B., He, J., Zhang, X., Zhang, Y. and Nenes, A.: Incorporation of advanced aerosol activation treatments into CESM/CAM5: Model evaluation and impacts on aerosol indirect effects, Atmospheric Chemistry and Physics, 14(14), 7485–7497, doi:10.5194/acp-14-7485-2014, 2014.

Ghan, S. J., Abdul-Razzak, H., Nenes, A., Ming, Y., Liu, X., Ovchinnikov, M., Shipway, B., Meskhidze, N., Xu, J. and Shi, X.: Droplet nucleation: Physically-based parameterizations and

comparative evaluation, Journal of Advances in Modeling Earth Systems, 3(10), M10001, doi:10.1029/2011MS00007, 2011.

Morales Betancourt, R. and Nenes, A.: Droplet activation parameterization: the population-splitting concept revisited, Geoscientific Model Development, 7(5), 2345–2357, doi:10.5194/gmd-7-2345-2014, 2014.

Partridge, D. G., Vrugt, J. a., Tunved, P., Ekman, a. M. L., Gorea, D. and Sorooshian, a.: Inverse modeling of cloud-aerosol interactions – Part 1: Detailed response surface analysis, Atmospheric Chemistry and Physics, 11(14), 7269–7287, doi:10.5194/acp-11-7269-2011, 2011.

Rothenberg, D. and Wang, C.: Metamodeling of Droplet Activation for Global Climate Models, Journal of the Atmospheric Sciences, 73(3), 1255–1272, doi:10.1175/JAS-D-15-0223.1, 2016.

West, R. E. L., Stier, P., Jones, A., Johnson, C. E., Mann, G. W., Bellouin, N., Partridge, D. G. and Kipling, Z.: The importance of vertical velocity variability for estimates of the indirect aerosol effects, Atmospheric Chemistry and Physics, 14(12), 6369–6393, doi:10.5194/acp-14-6369-2014, 2014.